# Unexpected large evasion fluxes of carbon dioxide from turbulent streams draining the world's mountains

Åsa Horgby [1], Pier Luigi Segatto [1], Enrico Bertuzzo [2], Ronny Lauerwald [3], Bernhard Lehner [4], Amber J. Ulseth [5], Torsten W. Vennemann [6] & Tom J. Battin [1]*

Inland waters, including streams and rivers, are active components of the global carbon cycle. Despite the large areal extent of the world's mountains, the role of mountain streams for global carbon fluxes remains elusive. Using recent insights from gas exchange in turbulent streams, we found that areal $CO_2$ evasion fluxes from mountain streams equal or exceed those reported from tropical and boreal streams, typically regarded as hotspots of aquatic carbon fluxes. At the regional scale of the Swiss Alps, we present evidence that emitted $CO_2$ derives from lithogenic and biogenic sources within the catchment and delivered by the groundwater to the streams. At a global scale, we estimate the $CO_2$ evasion from mountain streams to 167 ± 1.5 Tg C yr$^{-1}$, which is high given their relatively low areal contribution to the global stream and river networks. Our findings shed new light on mountain streams for global carbon fluxes.

[1] Stream Biofilm and Ecosystem Research Laboratory, École Polytechnique Fédérale de Lausanne (EPFL), Station 2, CH-1015 Lausanne, Switzerland. [2] Department of Environmental Sciences, Informatics and Statistics, University of Venice Ca' Foscari, 30123 Venezia Mestre, Italy. [3] Biogeochemistry and Earth System Modelling, Department of Geoscience, Environment and Society, Universite Libre de Bruxelles, 1050 Bruxelles, Belgium. [4] Department of Geography, McGill University, Montreal, Canada. [5] Department of Biological Sciences, Sam Houston State University, Huntsville, TX 77341, USA. [6] Institute of Earth Surface Dynamics, University of Lausanne, CH-1015 Lausanne, Switzerland. *email: tom.battin@epfl.ch

Since 2007, when a seminal publication[1] highlighted the relevance of inland waters for the global carbon cycle, estimates of $CO_2$ evasion fluxes from the world's streams, rivers, and lakes to the atmosphere have continuously moved upwards[2]. Current estimates of annual $CO_2$ evasion fluxes from inland waters are within the same range as ocean uptake fluxes of $CO_2$[3], although the fluxes are in the opposite direction. Streams and rivers alone are estimated to emit 650 Tg C yr$^{-1}$ (ref. [4]) to 1800 Tg C yr$^{-1}$ (ref. [5]) to the atmosphere, which is remarkable given that they contribute marginally to the Earth's non-glacierized land surface[6]. These fluxes are admittedly still poorly constrained, partly because of the lack of observations from various regions of the world and the poor quantification of stream networks, particularly their headwaters.

Mountains account for 25% of the Earth's land surface and the streams that drain them contribute more than a third to the global runoff[7]. Nevertheless, the role of mountain streams for global carbon fluxes has not yet been evaluated. To date, interest on $CO_2$ evasion fluxes has largely centered on streams and rivers draining low-altitude catchments in tropical[8,9] and boreal[10,11] regions. It is intuitive to assume that the lack of significant vegetation cover and soil carbon stocks in many mountain catchments, particularly in high-altitude catchments, have precluded research on carbon fluxes in the streams draining these systems. There are certainly exceptions to the inverse relationship between altitude and vegetation cover[12], such as the Paramo vegetation in the Andes, or more generally peatlands developing in high-altitude catchments. Furthermore, the lack of appropriate scaling relationships to predict the gas exchange velocity across the highly turbulent water surface of mountain streams has impeded the appreciation of their $CO_2$ evasion fluxes[13].

The few existing studies of $CO_2$ in mountain streams typically reveal low $pCO_2$ and occasionally even undersaturation relative to the atmospheric $pCO_2$ (e.g., ref. [14–17]). In line with this, temporally highly resolved measurements consistently indicate relatively low streamwater $pCO_2$ values (median: 397–673 µatm) throughout the year in twelve streams in the Swiss Alps (Supplementary Fig. 1, Supplementary Table 1). Not unexpected, these $pCO_2$ values are low compared to those measured in boreal[18] and tropical[19] headwaters, for instance, and would thus support the assumption that mountain streams contribute only marginally to global carbon fluxes. However, low $pCO_2$ in mountain streams can also result from high evasion fluxes, owing to elevated turbulence, compared to $CO_2$ supply from the catchment and $CO_2$ production from stream ecosystem respiration. This notion is in line with a recent study by Rocher-Ros and colleagues[20] showing low $CO_2$ concentrations in turbulent streams with high gas exchange velocity compared to a wide range of elevated $CO_2$ concentrations in low-turbulence streams with reduced gas exchange velocities and little supply limitation of $CO_2$.

In this study, we combine recent insights[13] into the gas exchange through the turbulent water surface of mountain streams with novel streamwater $CO_2$ concentration data to estimate $CO_2$ evasion fluxes from Swiss mountain streams, as well as from the mountain streams worldwide. We found unexpectedly high areal $CO_2$ evasion fluxes from these streams driven by high gas exchange velocities and a constant $CO_2$ supply from both biogenic and lithogenic sources. To our knowledge, this is the first large-scale attempt to estimate $CO_2$ evasion fluxes from mountain streams.

## Results and discussion

### Scaling relationships and parameter simulation.
To quantify $CO_2$ evasion fluxes, streamwater $CO_2$ concentration and exchange velocities must be estimated. Many current upscaling approaches involve aggregation of streamwater $pCO_2$, estimated from pH, DIC and alkalinity, into a single median value over very large regions (e.g., European Alps or Andes). This is then combined with gas exchange velocities at the stream or catchment scale[5,21]. While this approach has been often used for estimating regional and global $CO_2$, it might provide erroneous estimates[20]. We, therefore, opted for an alternative upscaling strategy involving similar spatial scales for streamwater $CO_2$ concentration and gas exchange velocity for each mountain stream individually.

We estimated streamwater $CO_2$ concentration from a linear regression model ($R^2 = 0.39$, $P < 0.001$) based on observations from 323 streams from the world's major mountain ranges (Methods; Supplementary Fig. 2). The streams included in the model drain catchments covering a broad range of lithologies, dominated by carbonate rocks (37%), siliciclastic sedimentary rocks (20%) and metamorphic rocks (20%). Furthermore, they cover similar mountain regions as those included in the Global River Chemistry database (GLORICH)[22] database and often used for upscaling[4,5] (Methods). Due to the low $pCO_2$ in mountain streams, we exclusively used measured $CO_2$ concentrations since $CO_2$ concentrations calculated from alkalinity, DIC and pH are prone to errors[5,23,24], which is the reason why they are often aggregated over larger regions. The model retained altitude (partial correlation: −0.65, $P < 0.001$), soil organic carbon content (partial correlation: 0.10, $P < 0.001$) and discharge (partial correlation: −0.09, $P < 0.001$) as predictors. Altitude affects streamwater $CO_2$ concentration along several lines. Streamwater temperature, terrestrial net ecosystem production (NEP)[12] and soil organic carbon content decrease with increasing altitude; NEP and soil organic carbon content are positively related to carbon fluxes in inland waters in general[25,26]. Besides elevation, discharge also scales broadly with channel slope, and bed roughness in mountain streams[13], all of them conducive to accelerated gas exchange and hence lower streamwater $CO_2$ concentration moving upstream.

We calculated the normalized gas exchange velocity $k_{600}$ (for $CO_2$ at 20 °C) using recently published scaling relationships based on energy dissipation ($eD$), which is the product of flow velocity, channel slope, and the gravity acceleration[13]. This relationship accounts for the high turbulence owing to steep-channel slopes and elevated streambed roughness of mountain streams. Channel width and flow velocity were calculated from hydraulic geometry scaling laws derived for mountain streams with an annual discharge smaller than 2.26 m$^3$ s$^{-1}$ (Methods; Supplementary Fig. 3). Channel slope was determined using streamlines combined with digital elevation models (DEM) (Methods). We acknowledge that this approach does not account for the step-pool structure in mountain streams that can locally increase channel slope[27]. Our slope estimates are therefore conservative (Methods). Moreover, we retained only streams with a predicted $eD$ smaller than 1.052 m$^2$ s$^{-3}$ to be within the boundary of the input data used for the gas exchange model (ref. [13]). In addition, we restricted the upper elevation boundary to 4938 m (a.s.l), corresponding to the highest sampling location included in our $CO_2$ model.

Rather than directly predicting streamwater temperature, channel width, flow velocity, $CO_2$ concentration, and temperature-dependent $CO_2$ exchange velocity ($k_{CO_2}$), we computed each of these parameters using Monte Carlo simulations with 10,000 iterations for each individual stream (Methods). Thereby we were able to propagate the error associated with each of these parameters into an uncertainty related to cumulative (e.g., regional or global) $CO_2$ evasion fluxes. We used the typology proposed by Meybeck and colleagues[7] for the identification of mountain catchments as those with an average altitude higher

than 500 m above sea level (a.s.l.) and an average relief roughness exceeding 20–40‰ depending on elevation as computed from digital elevation models (DEM) (Methods). A similar classification of mountains was also used to assess the relevance of mountains for water resources[28]. We then defined streams draining these regions as mountain streams.

**CO₂ evasion fluxes from Swiss mountain streams.** In a first step, we applied our upscaling approach to Switzerland where the availability of a high-resolution DEM (2 m) and accurate discharge data allowed us to reliably predict streamwater $CO_2$ concentrations and gas exchange velocity. Applying our selection criteria (i.e., restricting according to the mountain stream classifications, discharge and $eD$), we retained 23,343 streams (86% of them belonging to 1st to 4th Strahler order) for which we computed a median $k_{600}$ of 116 m d$^{-1}$ (7.5 and 650 m d$^{-1}$, 5th and 95th confidence interval quantiles, CI, respectively). The median of the corresponding temperature-corrected gas exchange velocities for $CO_2$ ($k_{CO_2}$) was 86.4 m d$^{-1}$ (CI: 6.0 and 462 m d$^{-1}$) (Fig. 1a). These numbers are higher than those used to calculate regional and global estimates of $CO_2$ evasion from streams and rivers[4,5]. We attribute this difference to the novel scaling relationships for $k_{600}$ (ref. [13]) that we used and that take into account the role of turbulence in accelerating gas exchange in mountain streams.

We estimate median streamwater $pCO_2$ of 705 µatm (CI: 380 and 1224 µatm) for the Swiss streams (Fig. 1b). By combining predicted streamwater $CO_2$ concentrations with $k_{CO_2}$ we compute a median areal $CO_2$ evasion flux of 3.5 kg C m$^{-2}$ yr$^{-1}$ (CI: −0.5 and 23.5 kg C m$^{-2}$ yr$^{-1}$) (Fig. 1c). These areal fluxes are unexpectedly high, equivalent or even higher than those reported for the Amazon[9,29] and boreal[10,30] streams, which, among the

inland waters, are typically considered as major emitters of $CO_2$ to the atmosphere. Over the 23,343 streams, these areal fluxes result in a total $CO_2$ evasion flux of 0.248 ± 0.012 Tg C yr$^{-1}$ from small Swiss mountain streams.

**Potential sources of CO₂ in Swiss mountain streams.** It is intuitive to assume that high evasion fluxes rapidly deplete $CO_2$ stocks in turbulent mountain streams and therefore cause the consistently low $pCO_2$ in these streams[20]. However, $pCO_2$ above saturation as often observed in mountain streams would imply a continuous supply of $CO_2$ able to sustain the high evasion fluxes. Groundwater is recognized as a potentially important delivery route of $CO_2$ into headwater streams[31–34]. To explore the potential of such $CO_2$ deliveries from groundwater into mountain streams in Switzerland, we applied a simple mass balance for $CO_2$ fluxes assuming that all $CO_2$ within a stream segment originates from groundwater discharge (Methods). Solving the mass balance for the groundwater $CO_2$ concentration in 3858 streams, we found that a median $CO_2$ concentration of 105 µmol L$^{-1}$ in the groundwater, equivalent to a median $pCO_2$ of 2195 µatm (CI: 42 and 38,867 µatm) would be required to sustain in principle the $CO_2$ evasion flux from these streams (Fig. 2a). This median value is indeed closely bracketed by measured $pCO_2$ (1343 to 4267 µatm) in the groundwater within two of our Swiss study catchments (Supplementary Table 2). Available data on groundwater $CO_2$ concentrations in mountain catchments are rare, and we therefore compare the expected groundwater $CO_2$ concentrations derived from our mass balance calculations also with data that are not necessarily from such catchments. For instance, maximum $pCO_2$ measured in groundwater in headwater catchments in Belgium, Czech Republic and Laos (Methods) were close to our expected 95th CI quantile of 51,647 µatm. Not unexpected, the variation of our estimates is large given the wide range of hydrological (e.g., fed by groundwater, snowmelt and glacier ice melt), geomorphological and geological characteristics of these streams and their catchments. Moreover, due to the lack of appropriate data, we were not able to include alkalinity as a potential sink for $CO_2$ in the mass balance[35]. Nevertheless, the agreement between estimated and reported $CO_2$ concentrations suggests that groundwater $CO_2$ contributions are potentially relevant to sustain the $CO_2$ evasion fluxes from mountain streams.

Our notion of external $CO_2$ sources to the mountain streams was further supported by various lines of geochemical evidence (Methods; Supplementary Note 1, Supplementary Figs. 4, 5). The streamwater ion balance suggests that the streams are representative for headwaters draining catchments with carbonate rocks[22,24] (Supplementary Fig. 4), and more important, that they are carbonate buffered to saturation (median calcite saturation index ranging from 2.48 to 4.11). This would imply a continuous supply of dissolved inorganic carbon (DIC) to the streams with new $CO_2$ re-equilibrating due to $CO_2$ evasion. As a further result, streamwater alkalinity was elevated (median: 2.05 meq L$^{-1}$; range: 0.94 to 2.85 meq L$^{-1}$), even beyond the threshold where DIC from carbonate weathering can drive $CO_2$ supersaturation in numerous lakes worldwide[36]. Therefore, in conjunction with respiratory $CO_2$ from soils, carbonate minerals can be a potential source to the $CO_2$ evasion flux from the mountain streams.

The notion of a lithogenic $CO_2$ source is further supported by stable isotope analyses of streamwater DIC (Methods). Across our study streams ($n = 134$), we found δ$^{13}$C values ranging from −11.6 to −1.76‰ VPDB (median: −5.8, CI: −9.9 and −2.5) (Fig. 2b, Supplementary Fig. 5). Overall, these values are closely bracketed by reported isotopic compositions for soil organic matter (ranging from −30 to −24[37]) and carbonate rocks (close

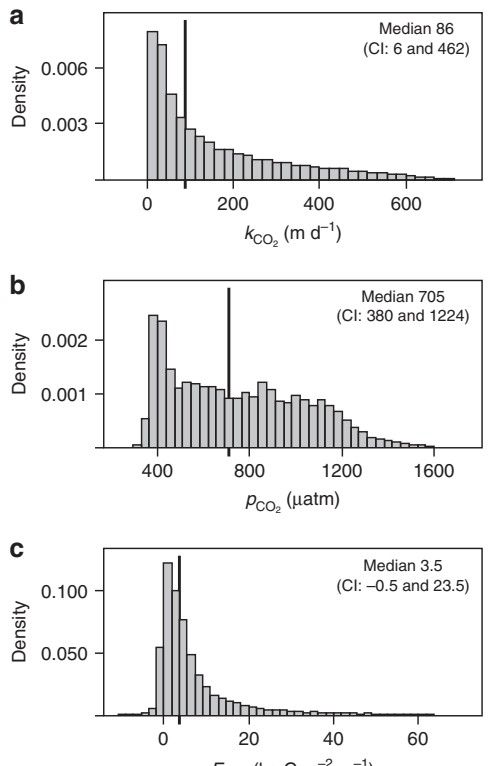

**Fig. 1** Patterns of $CO_2$ in streams in the Swiss Alps. The distributions of $k_{CO_2}$ (**a**), $pCO_2$ (**b**) and areal $CO_2$ fluxes ($F_{CO_2}$) (**c**) for 23,343 mountain streams in Switzerland

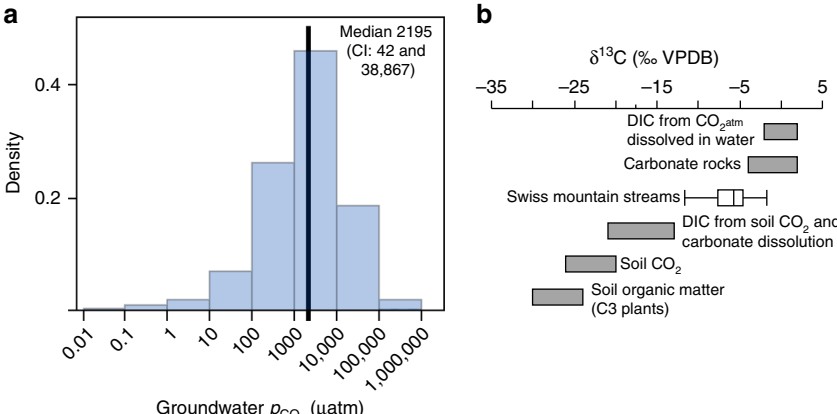

**Fig. 2** Sources of $CO_2$ in streams in the Swiss Alps. **a** Estimated groundwater $pCO_2$ in Swiss mountain catchments. **b** Isotopic compositions of dissolved inorganic carbon ($\delta^{13}$C-DIC) indicates biogenic (e.g., soil respiration) as well as geogenic sources (more enriched) (‰ Vienna Pee Dee Belemnite, VPDB). End-members are adopted from refs. [37,38]. The box plot shows median and quartile $\delta^{13}$C-DIC compositions repeatedly sampled across the 12 Swiss sites ($n = 134$; 7–15 samples per stream) (calculated in JMP 13, SAS Institute Inc., USA)

to zero[37]) as two end-members of the $\delta^{13}$C variability continuum[37,38]. This implies contributions from both the respiration of organic carbon and lithogenic sources to the streamwater DIC pool. Given the overall very low concentrations of dissolved organic carbon (DOC; $254 \pm 124\,\mu$g C liter$^{-1}$; Methods) in our study streams, we suggest that most of the depleted DIC is from respiratory $CO_2$ from soils and delivered by groundwater to the streams. The delivery of DIC from lithogenic sources (mostly carbonate weathering) into streams and its subsequent outgassing as $CO_2$ into the atmosphere is increasingly being recognized[33,35,39]. However, the underlying processes seem less evident and certainly require more attention in the future. We suggest that depending on the carbonate buffering capacity, both dissolution of atmospheric $CO_2$ (but also from soil respiration) could lower the pH in the soil water, groundwater and ultimately in the streamwater. If the streamwater is already saturated in $CO_2$ with respect to the atmosphere, DIC would be converted into $CO_2$ that may ultimately outgas from the stream[35]. Furthermore, cold water can dissolve more $CO_2$, which facilitates the dissolution of carbonates in the soil water and groundwater; if these waters warm in the stream, carbonates can re-precipitate with the concurrent release of $CO_2$ (ref. [40]). We suggest that this retrograde solubility further adds to the $CO_2$ outgassing from streams when colder groundwater transports dissolved carbonates to warmer streamwater in summer. Whereas the relative effect of pH changes on streamwater $pCO_2$ may outweigh the effects of temperature, we suggest that their combination can be important for the conversion of bicarbonates to carbonic acid (and $CO_2$) in mountain streams.

**$CO_2$ evasion fluxes from the world's mountain streams.** In a second step, we extrapolated our findings from the Swiss Alps to assess the $CO_2$ evasion fluxes from the world's mountain streams. The accuracy of geomorphological and hydrological parameters extracted from DEMs and other maps depends on their spatial resolution. Therefore, before transferring our approach from the Swiss streams, we compared the statistical distributions of elevation, stream slope, discharge obtained from our high-resolution dataset with those obtained from low-resolution data available for approaches at the global scale (Supplementary Note 2). We found surprisingly good agreement between both approaches (Supplementary Fig. 6), and were therefore confident to proceed with the upscaling of $CO_2$ fluxes from mountain streams at the global scale. Here we used the Global River Classification (GloRiC)

database[41], an extended version of HydroSHEDS, that describes drainage networks of Earth's surface in 15 arc-second (~500 m) spatial resolution including the networks above the 60°N latitude. These northern regions were poorly represented in previous estimates of global $CO_2$ evasion fluxes from streams and rivers[4,5]. Discharge data included in the GloRiC database were used to infer stream flow velocity and channel width (Methods). Rather than presenting streamwater $pCO_2$, we present the $CO_2$ gradient ($\Delta CO_2$) as the difference between streamwater and atmospheric $CO_2$ concentration. In combination with $k_{CO_2}$, $\Delta CO_2$ is useful to understand the drivers of the $CO_2$ fluxes and to evaluate the spatial distribution of potential sources and sinks of $CO_2$.

Using the same selection criteria as for the Swiss mountain streams, we retained a total of 1,872,874 stream segments for which we calculated a global median $k_{600}$ of 31.4 m d$^{-1}$ (CI: 4.6 and 460 m d$^{-1}$) and a corresponding median $k_{CO_2}$ of 25.6 m d$^{-1}$ (CI: 3.5 and 411 m d$^{-1}$) (Supplementary Fig. 7A). These are 3.7 and 3.4 times lower, respectively than the average gas exchange velocity calculated for the Swiss streams. The skewed distribution of global $k_{CO_2}$ towards smaller values may result from the abundant streams draining large plateaus (e.g., interior Tibetan Plateau, Altiplano) (Supplementary Fig. 7A). We predicted a median streamwater $pCO_2$ of 737 μatm (CI: 317 and 1644 μatm) (Supplementary Fig. 7B), which are lower than the global predictions of 2400 to 3100 μatm from studies that were likely biased towards larger streams and rivers[4,5]. However, our values are comparable with $pCO_2$ values reported from streams that drain mountain regions[4,17] and that were not included in our predictive model for streamwater $CO_2$ concentration. Overall, this agreement corroborates our $CO_2$ model and Monte Carlo simulation approach. We calculated a median global areal $CO_2$ evasion flux of 1.1 kg C m$^{-2}$ yr$^{-1}$ (CI: $-0.54$ and 32 kg C m$^{-2}$ yr$^{-1}$) (Supplementary Fig. 7C). Overall, we found negative $CO_2$ fluxes in 10.8% of the streams (i.e., these streams are potential sinks of atmospheric $CO_2$).

Overall, the spatial distribution of $k_{CO_2}$, $\Delta CO_2$, and areal $CO_2$ fluxes followed the variation of mountain topology (Fig. 3). For instance, streams (median elevation: 4236 m a.s.l.; CI: 2676 and 4886 m a.s.l.) draining the inner Tibetan Plateau have low $pCO_2$ (median: 288 μatm; CI: 194 and 449 μatm) translating into a negative median $\Delta CO_2$ of $-56$ mg C m$^{-3}$ (CI: $-105$ and 23 mg C m$^{-3}$). Similar $CO_2$ concentrations close to equilibrium were also reported by others for streams[42] and lakes[43] on the Tibetan Plateau. These gradients result in an overall negative areal $CO_2$

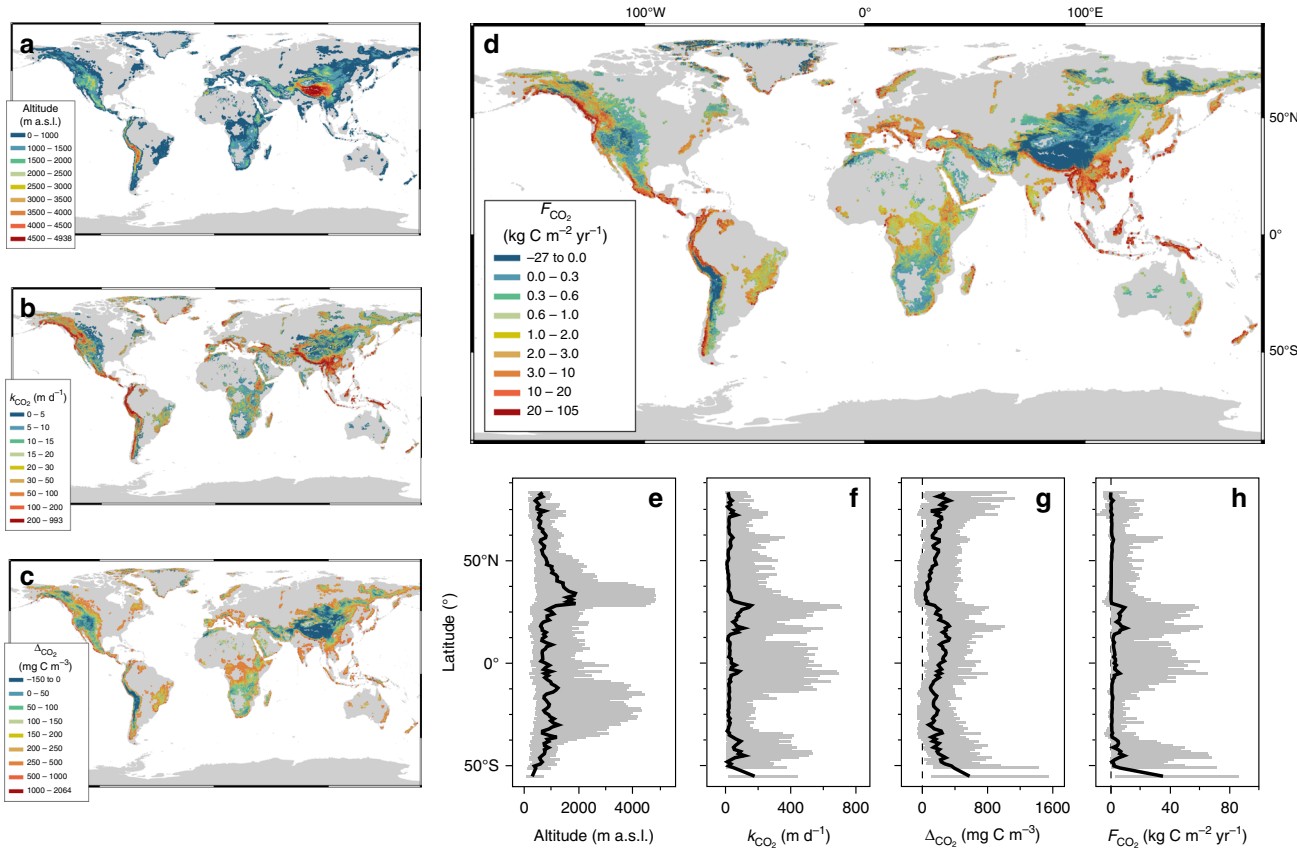

**Fig. 3** Global distributions of $CO_2$ in mountain streams. **a** Altitude of mountain streams, where mountains as defined according to ref. [7]. **b–d** Global distribution of predicted $CO_2$ exchange velocities ($k_{CO_2}$), $CO_2$ gradients between the streamwater and the atmosphere ($\Delta CO_2$) and the areal $CO_2$ fluxes ($F_{CO_2}$), respectively. **e–h** Latitudinal transects of these same parameters at 1-degree resolution (shown are median values in black and 5 and 95% confidence intervals in gray)

flux of $-0.36$ kg C m$^{-2}$ yr$^{-1}$ (CI: $-4.29$ and $0.87$ kg C m$^{-2}$ yr$^{-1}$) (Supplementary Fig. 8). Our estimates would, therefore, suggest that the Tibetan Plateau streams potentially act as a net sink (total flux: $-1.46$ Tg C yr$^{-1}$; CI: $-1.52$ and $-1.39$ Tg C yr$^{-1}$) of atmospheric $CO_2$. On the other hand, tropical mountain streams generally exhibited higher $\Delta CO_2$ values, likely due to terrestrial inputs of $CO_2$ from soil respiration[19] and the in-stream degradation of terrestrial plant material[44]. At higher elevations, outside the tropical biome, lower $\Delta CO_2$ values were compensated by high $k_{CO_2}$ because of steep stream channels, which resulted in high areal $CO_2$ fluxes.

We estimated the net global $CO_2$ evasion flux from mountain streams by cumulating the average positive and negative $CO_2$ fluxes calculated from Monte Carlo simulations (10,000 iterations) for 1,872,874 streams (Methods). We obtained a net global $CO_2$ evasion flux of 166.6 Tg C yr$^{-1}$ (CI: 165.9 and 167.4 Tg C yr$^{-1}$) from mountain streams. The magnitude of this evasion flux is high given that the mountain streams included in this study cover a surface area of 34,979 km$^2$, which corresponds to 4.5% or 6.0% of the global extent of streams and rivers as recently published by ref. [6] (773,000 km$^2$) and as calculated from GloRiC (587,630 km$^2$), respectively (Methods). Our estimate of the global net $CO_2$ evasion flux from mountain streams is within the same range as the total $CO_2$ evasion fluxes from tropical streams (excluding the large rivers and their floodplains) (160–470 Tg C yr$^{-1}$)[4,5,9] and substantially higher than those reported from the boreal-arctic streams and rivers (14–40 Tg C yr$^{-1}$)[4,45].

As for the Swiss Alps, we suggest that, in the absence of major soil development within the catchment, DIC derived from carbonates may contribute in conjunction with soil respiratory $CO_2$ to the outgassing from these streams. This assumption would be supported by the fact that many of the world's streams drain catchments containing carbonate rocks[24,46,47] and that many of them have an alkalinity (median: 1.51 meq L$^{-1}$; CI: 0.09, 5.13 meq L$^{-1}$; from GLORICH[22]) that is relevant for DIC from carbonate dissolution to drive $CO_2$ supersaturation[36]. Our findings thus contribute to increasing understanding that $CO_2$ from carbonate dissolution plays a hitherto poorly recognized role for the $CO_2$ evasion fluxes from inland waters[32,33,36]. Alternatively, the oxidation of rock-bound carbon (i.e., petrogenic carbon) can be a source of $CO_2$[48], especially in glacierized catchments regularly exposed to frost shattering[49]. This is often the case with mountain streams.

Therefore, we propose that groundwater deliveries of geogenic and hence ancient $CO_2$, besides the $CO_2$ from soil respiration, is a significant contributor to the $CO_2$ efflux from mountain streams. This would be facilitated by topographic roughness of mountain regions generating longer groundwater flow pathways and by bedrock fractures enhancing permeability and deep infiltration, and ultimately resulting in longer residence times of water within mountain catchments[50]. Deeper infiltration and extended residence times of groundwater would also increase the concentration of weathering products[51] in the groundwater that enters mountain streams.

**Temporal variations.** The extrapolation of $CO_2$ fluxes from streams and rivers to a regional or global scale rarely takes into account the temporal variability of the fluxes[4,5]. This runs against

                    

the recognition that $CO_2$ fluxes from streams and rivers can change on a seasonal and diurnal basis[16,52]. Furthermore, depending on the zero degree isotherm[53], mountain streams can fall dry, or they are snow-covered during winter. Factoring this variability into an upscaling effort of gas fluxes is difficult though exposition, terrain slope and groundwater upwelling all are factors that affect the snow cover locally. For instance, even during winter mountain streams can have reaches without snow cover, which serve as hotspots for outgassing of $CO_2$ that has accumulated upstream from groundwater deliveries into the snow-covered channel.

To assess the potential inaccuracy emanating from the temporal variation of $CO_2$ fluxes for our upscaling, we compared the median $CO_2$ flux (on an annual basis) calculated from the continuous measurements (every 10 min) with the predicted annual $CO_2$ flux in several of our Swiss study streams with rather complete time series (Supplementary Note 3). We found good congruence between the measured and predicted fluxes ($R^2 = 0.68$, $P = 0.02$, slope = $0.93 \pm 0.28$), which we consider as a further proof of the robustness of our scaling approach (Supplementary Fig. 9).

**Uncertainties and limitations**. Upscaling $CO_2$ evasion fluxes from streams and rivers is not an easy task and requires an element of simplification and speculation. This is particularly true for small mountain streams. A first level of uncertainty emanates from the definition of a mountain and the spatial resolution and aggregation used to identify mountain regions. We used the parsimonious aggregation approach at a 0.5° spatial resolution as previously done to quantify runoff in mountain regions[7,28]. We recognize that applying the filters (e.g., spatial resolution, relief, altitude) differently may lead to different global maps of mountain streams[7,28].

Channel width is inherently difficult to estimate for small streams. Rather estimating channel width from hydromorphological scaling relationships that also require information on hydraulic resistance[6,54], we derived channel width from hydraulic scaling relationships specifically established for mountain streams in combination with discharge from the GloRiC database[41]. Discharge is available at the level of spatial resolution required for upscaling $CO_2$ fluxes, whereas parameters for hydraulic resistance are not. Furthermore, by using discharge to infer width and velocity, but also to predict streamwater $CO_2$ concentration, we constrain errors to the same source. Our approach yielded a minimum stream channel width of 0.32 m, which is identical with the stream width reported by Allen and colleagues[54] as the characteristic most abundant stream width in headwater catchment.

The overall uncertainty associated with our regional and global $CO_2$ fluxes appears small compared to previous upscaling studies[4,5]. This is inherent to the structure of our uncertainty computation that assumes that errors in the estimation of fluxes at the stream segment level are independent. Therefore, summing up largely uncertain stream segment fluxes results in a global estimate with a small uncertainty compared to the median value, because errors average out if they are independent[55]. This is analogous to the reduction of the coefficient of variation of the sum of identically distributed, independent random variables, as predicted by the central limit theorem. Assuming that errors are fully independent is an approximation, of course, as is the assumption of fully correlated error as the opposite extreme. Therefore, we also computed the uncertainty with the latter assumption (Methods) and found a larger uncertainty associated with the total flux for the mountain streams in Switzerland (CI: $-0.107$ and $0.939\,\mathrm{Tg\,C\,yr^{-1}}$) and worldwide (CI: $-27.7$ to

$561.9\,\mathrm{Tg\,C\,yr^{-1}}$). The large discrepancy between the two uncertainty approaches is not unexpected and the real uncertainty is probably somewhere between both approaches.

In summary, our study reveals small streams of the world's mountains as an important yet hitherto poorly appreciated component of the global carbon cycle. High turbulence, induced by elevated channel slopes and streambed roughness, accelerates the evasion of $CO_2$ delivered from geogenic and biogenic sources by the groundwater into the mountain streams. The proper integration of the $CO_2$ evasion from mountain streams will further reduce the uncertainties around global carbon fluxes in inland waters.

## Methods

**On-line measurement of $pCO_2$ in Swiss streams**. We operated 12 sensor stations in high-altitude Alpine catchments; 4 catchments with 3 stations in each (Supplementary Fig. 1). Site elevation ranges from 1200 to 2161 m a.s.l., stream slope from 0.033 to 0.160 m m⁻¹ and annual mean discharge from 0.02–2.26 m³ s⁻¹. At the stations, we measured streamwater $pCO_2$ continuously (10 min intervals) during two years (2016–2018) (Supplementary Table 1). Prior to deployment, we prepared the $pCO_2$ sensors (Vaisala CARBOCAP® Carbon Dioxide Transmitter Series, GMT220, Finland) with a porous polytetrafluoroethylene (ePDFE) semi-permeable membrane that we sealed with liquid electrical tape[56]. We protected our water-proof $pCO_2$ sensors with fine-grained mash, PVC tube, and metal casing. We connected the sensors to two 12-volt batteries in series coupled with solar panels located at the streambed side.

**Geochemical analyses and potential $CO_2$ sources**. Filtered streamwater samples (Mixed Cellulose Ester filter, 0.22 µm) were repeatedly collected for the analyses of cation and anion concentrations between 2016 and 2018 in twelve study streams in the Swiss Alps and analyzed using ion chromatography (ICS-3000 Dionex, Sunnyvale, CA, USA). We also sampled streamwater for dissolved organic carbon (DOC) concentration. For DOC, we filtered (GF/F filters, Whatman) streamwater into 40 mL acid-washed and pre-combusted glass vials and analyzed within 1–3 days (Sievers M5310c TOC Analyzer, GE Analytical Instruments, USA). The accuracy of the instrument is ±2%, precision <1% and detection limit 1.83 µmol C L⁻¹.

Furthermore, we measured concentrations and the isotopic composition of dissolved inorganic carbon (DIC; δ¹³C-DIC). Samples for DIC concentration and δ¹³C-DIC were collected in 12 mL glass vials and filtered (Mixed Cellulose Ester filter, 0.22 µm) to retain the dissolved fraction. In the laboratory, we injected 2 mL streamwater into pre-flushed (synthetic air, $pCO_2 < 5$ ppm) exetainers containing 300 µL of 85% orthophosphoric acid. Samples were then shaken (2 min) and equilibrated overnight at room temperature. DIC samples were analyzed on a G2201-$I$ Picarro Instrument (Santa Clara, CA, USA) as $CO_2$ released from the reaction with orthophosphoric acid. There are three possible sources of DIC: atmospheric $CO_2$, weathered carbonates, and soil-derived respired $CO_2$. Weathering and atmospheric exchange enriches the DIC stable isotope signature[57,58] where atmospheric $CO_2$ and rock carbonate will largely overlap in their δ¹³C-DIC value if the rock is originally of marine origin. In contrary, contributions from respiration deplete the isotopic signature, depending on the plant type and diagenetic state of the decomposed organic matter[37,38].

**Stream hydraulic geometry**. We established hydraulic geometry scaling relationships from mountain streams in the Swiss Alps (Supplementary Fig. 1), where we derived annual mean stream channel width ($w$), depth ($z$) and flow velocity ($v$) from annual mean discharge (Q) as follows (Supplementary Fig. 3).

$$w = 7.104 \times Q^{0.447} \tag{1}$$

$$z = 0.298 \times Q^{0.222} \tag{2}$$

$$v = 0.668 \times Q^{0.365} \tag{3}$$

We performed a total of 141 slug releases where we added sodium chloride (NaCl) at the top of each reach (in average 12 slugs per site) and measured the change in specific conductivity at the bottom of the reaches. By measuring the change in specific conductivity, which we converted to mass by applying a pre-established relationship between specific conductivity and the conductivity potential of the added NaCl, we estimated discharge. We also estimated the travel time as the time for the NaCl to reach the bottom of the reach (i.e., the peak in the specific conductivity). To obtain average flow velocity we divided reach length by the travel time. We also measured stream width and stream depth.

In comparison to previous scaling relationships[59], our relationships are more representative for mountain streams, where steeper slopes induce higher flow velocities and narrower channels. Annual mean discharges ranged from 0.02 to 2.26 m³ s⁻¹ in our study streams ($n = 12$) in the Swiss Alps. The maximum annual mean discharge was used as an upper boundary within which we consider our

hydraulic geometry scaling valid, and we, therefore, restricted our data for all further analyses to streams with maximal annual mean discharge of 2.26 m³ s⁻¹. Hence we restrained our definition of mountain streams further and consider our estimates of $CO_2$ fluxes from mountain streams as conservative as we discarded streams with $Q > 2.26$ m³ s⁻¹.

**$CO_2$ flux calculations**. We estimated the gas transfer velocity ($k_{600}$, m d⁻¹) using the following piece-wise power-law relationships as recently published by Ulseth and colleagues

$$\ln(k_{600}) \, for \, eD > 0.02 = 1.18 \times \ln(eD) + 6.43 \quad (4)$$

$$\ln(k_{600}) \, for \, eD < 0.02 = 0.35 \times \ln(eD) + 3.10 \quad (5)$$

where $eD$ is the stream energy dissipation rate, which is the product of slope, flow velocity and the gravity acceleration. In order to use this gas transfer velocity equation, we restricted the streams used for our analyses to those where $eD$ did not exceed 1.052 m² s⁻³, which was the maximum $eD$ used in scaling relationship by Ulseth and colleagues[4].

To convert $k_{600}$ into $k_{CO_2}$, we calculated $CO_2$ saturation ([$CO_{2sat}$] as

$$[CO_{2sat}] = 400.40 \times \frac{P_{atm}}{P_{std}} \times KH \quad (6)$$

using annual mean atmospheric $CO_2$ in 2017 (400.40 µatm) measured at Jungfraujoch, Switzerland (World Data Centre for Greenhouse Gases (WDCGG), Japan, 2018). Then, by multiplying with the Henry constant ($K_{H,}$ mol L⁻¹ atm⁻¹) and the ratio between atmospheric pressure ($P_{atm}$, atm) and standard pressure of 1 atmosphere ($P_{std}$, atm) we calculated the $CO_2$ saturation ([$CO_{2sat}$], mol L⁻¹).

In Eq. (7), $P_{atm}$ changes with elevation ($E$),

$$P_{atm} = P_0 \times \frac{T_b}{T_b + \lambda \times E}^{\frac{g \times m}{R \times \lambda}} \quad (7)$$

where $P_0$ is the International standard atmosphere (ISA) values of sea level pressure (101,325 Pa) and $T_b$ is an assumed sea level temperature of 19 °C (292.15 K). λ is the temperature lapse rate (−0.0065 K m⁻¹), $g$ is the gravity acceleration (9.80616 m s⁻²), m is the molecular weight of dry air (0.02897 kg mol⁻¹), and $R$ is the gas constant (8.3143 J mol⁻¹ K⁻¹). We multiplied the values derived from Eq. (7) with $9.86923 \times 10^{-6}$ to obtain $P_{atm}$ in atmospheres[60].

$K_H$ is a function of water temperature ($T_K$, Kelvin), where A (108.3865), B (0.01985076), C (−6919.53), D (−40.4515) and E (669365) are constants[61].

$$K_H = 10^{A + B \times (T_K) + \frac{C}{T_K} + D \times log10(T_K) + \frac{E}{T_K^2}} \quad (8)$$

To estimate streamwater temperature, we extracted gridded air temperatures[62], which we translated into streamwater temperatures according to a relationship between streamwater temperature ($T_w$) and air temperature ($T_{air}$)[4].

$$T_w = 3.941 \pm 0.007 + 0.818 \pm 0.0004 \times T_{air} \quad (9)$$

We used the temperature-dependent Schmidt scaling (10)[63] to convert $k_{600}$ (Eqs. (4), (5) respectively) to $k_{CO_2}$ (11).

$$Sc_{CO_2} = 1923.6 - 125.06 \times T_w + 4.3773 \times T_w^2 - 0.085681 \times T_w^3 + 0.00070284 \times T_w^4 \quad (10)$$

$$k_{CO_2} = \frac{k_{600}}{\left(\frac{600}{Sc_{CO_2}}\right)^{-0.5}} \quad (11)$$

To estimate streamwater $CO_2$, we collected data from Swiss Alpine streams, which we combined with stream data from Austria[16], Kenya[64], USA (A. Agerich, personal communication, Kuhn et al., 2017, C. Kuhn, personal communication; P. del Giorgio, personal communication; P. Raymond, personal communication), Brazil[65], Tibet and China (refs. [15,66], L. Ran personal communication), and New Zealand (V. De Staercke; M. Styllas; M. Tolsano, personal communication). We restricted our dataset to only encompass mountain streams[7] with annual mean discharges[41] below 2.26 m³ s⁻¹. We predicted streamwater $CO_2$ concentration from a linear regression model using mean channel elevation ($E$)[67,68], mean annual discharge ($Q$)[41] and soil organic carbon content (SOC, g kg⁻¹)[69] (Supplementary Fig. 2), that we extracted with QGIS using the Point sampling tool. The model is based on a collection of 323 direct measurements of streamwater $CO_2$ concentration from mountain streams that were selected according to our selection criteria (i.e., elevation, relief, discharge). The regression model

$$\ln(CO_2) = -0.647 \pm 0.052 \times \ln(E) - 0.094 \pm 0.014 \times \ln(Q) \\ + 0.099 \pm 0.029 \times \ln(SOC) + 7.287 \pm 0.427 \quad (12)$$

explained 39% of the variation ($R^2 = 0.39$, $n = 323$, $p < 0.0001$) in streamwater $CO_2$ concentration.

Finally, areal $CO_2$ fluxes (g C m⁻² d⁻¹) were calculated as

$$F_{CO_2} = k_{CO_2} \times \Delta_{CO_2} \quad (13)$$

where the $CO_2$ gradient $\Delta CO_2$ (converted to g C m⁻³) is the $CO_2$ gradient between

the streamwater and the atmosphere. To estimate total fluxes, we first estimated stream area ($A$) from stream width as derived from the hydraulic scaling relationships (1) and stream length ($L$) defined in the stream network dataset for Swiss (Federal Office for the Environment (FOEN) Switzerland, 2013) and global[41] streams. The total $CO_2$ flux per stream was then calculated as areal $CO_2$ fluxes multiplied with stream area.

**Monte Carlo simulations and uncertainties**. We used Monte Carlo (Matlab 2017b) approaches to simulate the parameters (i.e., streamwater $CO_2$ concentration, channel width, streamwater temperature, flow velocity and $k_{CO_2}$) required for the calculation of $CO_2$ evasion fluxes and to estimate related uncertainties for each individual stream. We used two different approaches to quantify the uncertainty. A first approach was based on the assumption that errors in the calculation of $F_{CO_2}$ for each stream were independent. For each stream and for each of the 10,000 iterations, we perturbed the various scaling relationships by randomly extracting error approximations from their corresponding residual probability distribution. We thereby created for each Monte Carlo simulation a random extraction of the streamwater $CO_2$ concentration, stream width, streamwater temperature, flow velocity and $k_{CO_2}$ values for all streams, and finally 10,000 estimates of areal $CO_2$ evasion fluxes (according to (13)). We classified the upper 99.5 percentiles of all slope, streamwater $CO_2$ and areal $CO_2$ flux estimates as outliers and removed them from further analysis, to avoid unrealistically inflated values. Then, for each iteration, we derived a total flux by summing up the fluxes from all streams accounting for their contributing area. We thereby obtained 10,000 total flux estimates, from which we extracted the mean $CO_2$ evasion flux as well as the 5th and 95th percentiles as confidence intervals. For this approach, the largest uncertainty was related to the $k_{600}$ model, while the hydraulic scaling relationships (for flow velocity and width), the streamwater temperature and streamwater $CO_2$ model contributed less to the overall uncertainty. The streamwater $CO_2$ concentrations, $k_{CO_2}$, and areal $CO_2$ fluxes reported in our study refer to the means obtained from the 10,000 iterations. The propagated $CO_2$ fluxes were summed to obtain a total estimate of the annual $CO_2$ evasion flux. As a consequence, the errors introduced at the different iterations average out. This resulted in a narrow $CO_2$ flux distribution due to the assumption of independent errors.

A second approach was based on the assumption that all errors in the calculation of $F_{CO_2}$ for each stream were perfectly dependent on each other. Thus, instead of summing the $F_{CO_2}$ across all streams and then draw the distribution from the different iterations, we used the distributions derived for each stream from the Monte Carlo simulations, from which we calculated the mean and confidence intervals. Then, we summed all means and confidence intervals separately to obtain the total $F_{CO_2}$ estimate and the uncertainties. With this approach, we obtained much larger uncertainties compared to the first approach. Because, under the assumption of error dependency, the percentiles of the total $F_{CO_2}$ distribution equal the sum of the percentiles of the single stream distributions. Reality is probably somewhere in between the two approaches and we, therefore, decided to report uncertainties estimated with both approaches.

**Definition of mountain streams**. We defined mountain streams as those draining terrain with an elevation above 500 m a.s.l. and more than 20 to 40‰ in relief roughness depending on elevation[7]. This approach was previously used to estimate water resources and runoff from the world's mountains[7,28]. We used the Global Multi-resolution Terrain Elevation Data (GMTED2010)[67], which we aggregated to 0.5° using mean elevations (ArcGIS 10.5, Aggregate tool). We derived relief roughness from the DEM (QGIS 3.2.1. with GRASS 7.4.1, Roughness tool) where relief roughness was calculated as the difference in a pixel's maximum and minimum elevation divided by half the pixel length.

**Groundwater $CO_2$ mass balance**. We calculated the groundwater $CO_2$ concentration that would be required in principle to sustain the $CO_2$ evasion fluxes from 3858 mountain streams in the Swiss Alps. To do so, we first estimated the flow between stream segments (From/To Node tool, Arc Hydro, Esri 2011). Then, we established a mass balance similar to refs. [21,32], where the difference in discharge ($Q$, m³ s⁻¹) between two stream segments, $x$ and $x + 1$, is assumed to be due to groundwater inflow ($Q_{GW}$). Therefore, the groundwater $CO_2$ concentration ($C_{GW}$, µmol m⁻³) can be calculated as;

$$C_{GW} = \frac{f_x + (CQ)_{x+1} - (CQ)_x}{Q_{GW}} \quad (14)$$

where $f_x$ is the $CO_2$ evasion flux (µmol s⁻¹), and $C$ (µmol m⁻³) is the $CO_2$ concentration in the streamwater.

Groundwater mass balance indicated that a median groundwater $pCO_2$ of 2195 µatm (CI: 42 and 38,867 µatm) would be required to sustain the $CO_2$ evasion flux from Swiss streams (computed for $n = 3858$). We compared the results obtained from the groundwater mass balance with groundwater $pCO_2$ data sampled in two of our study catchments; catchment B (Supplementary Fig. 1) had a median groundwater $pCO_2$ of 1343 µatm (CI: 245 and 1936 µatm, $n = 9$; Supplementary Table 2) and catchment C had a median groundwater $pCO_2$ of 4267 µatm (CI: 2230 and 6303 µatm, $n = 2$; Supplementary Table 2). Yet, those few measurements of

groundwater $p$CO$_2$ may underestimate groundwater CO$_2$ concentrations; measurements of groundwater $p$CO$_2$ in Belgium[70], Laos and Czech Republic (C. Duvert, personal communication) are 10-fold higher with measured values up to almost 50,000 μatm (highest measured value: 47,374).

**Extrapolating CO$_2$ evasion.** For Switzerland, we used the stream network from the Swiss Federal Office for the Environment (FOEN, 2013), which we combined with mean simulated natural annual discharge data (1981–2000) (FOEN, 2016). We created a node layer (Node tool) in QGIS and we extracted elevation data (Point sampling tool) from a highly resolved (2 m) digital elevation model (DEM) (Geodata © swisstopo). Prior to sampling, we resampled (nearest neighbor, median, 3-pixel radius in SAGA GIS 2.3.2) the DEM to remove outliers. We calculated stream slopes (Matlab 2017b) as the elevation difference per stream divided by the predefined stream length (FOEN, 2013). We extracted SOC content[69] for every node, which we averaged to mean values per stream. Similarly, we extracted monthly air temperatures[62], which we averaged over the year and converted to streamwater temperature[4] (9).

We estimated global CO$_2$ fluxes from mountain streams using a similar approach as for the Swiss streams. We used the GloRiC stream network at 15 arc-seconds (~500 m) spatial resolution, including streams north of 60° latitude[41]. To estimate stream channel slopes, we first resampled the DEMs to remove outliers (nearest neighbor, median value in a 3-pixel radius) in SAGA GIS 2.3.2. We used the SRTM 90 m[68], which we combined with the 30 s GMTED elevation layer[67] for streams above 60°N. We created a node layer from the GloRiC stream network from which we extracted elevation (Node tool, QGIS) and calculated stream gradients (Matlab 2017b) as the elevation difference per stream divided by the predefined stream length[41]. We used the discharge from Dallaire and colleagues[41]. For every node, we also extracted the geopredictors required in the CO$_2$ model; soil organic carbon content[69] and air temperature[62] which we converted to streamwater temperature[4].

To approximate the total global stream area, we used all the streams and discharge data included in the GloRiC dataset[41] and inferred width from the hydraulic scaling relationship equations for larger rivers[59]. In other words, since we wanted an approximate estimate for all streams and rivers, we used a well-established hydraulic scaling relationship (ref. [59]) to estimate stream width, which we combined with stream lengths from the GloRiC dataset. We summed all stream areas to estimate a total stream surface area (to estimate the stream area of mountain streams we used our own scaling hydraulic relationship to obtain stream width).

## Data availability

Data from the 12 Swiss sites presented in this study, as well as codes and data required for the uncertainty analyses, can be found at https://doi.org/10.6084/m9.figshare.9925097.v1.

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

## Acknowledgements

We are grateful to Alba Algerich, James Thornton, Paul del Giorgio, Andrea Popp, Anne Marx, Clément Duvert, Catherine Kuhn, Lishan Ran, Steven Bouillon, Bin Qu, Vincent De Staercke, Michail Styllas, Matteo Tolsano and Peter Raymond for sharing their data with us. We acknowledge the help of Nicolas Escoffier, Valentin Sahli, Rémy Romanens, Félicie Hammer, Amin Niayifar and Marta Boix Canadell with fieldwork. Peter Raymond provided useful comments on an earlier version of the paper. Financial support came from the European Union's Horizon 2020 research and innovation programme under the Marie Sklodowska-Curie grant agreement No 643052 (C-CASCADES project) and the Swiss Science Foundation (SNF, 200021_163015) to T.J.B.

## Author contributions

Å.H. and T.J.B. conceptualized the study; Å.H. acquired and analyzed the data with the support from P.L.S. as well as E.B., R.L., B.L., A.J.U., and T.W.V; T.J.B. wrote the paper with the help of Å.H. and with contributions from all other authors.

## Competing interests

The authors declare no competing interests.
