## [Peer Review File · Nature Communications]

Reviewers' comments:

Reviewer #1 (Remarks to the Author):

Review of Horgby et al: Unexpected large evasion fluxes of carbon dioxide from turbulent streams draining the world's mountains.

The manuscript by Horgby et al. assesses the importance of mountainous streams and rivers in terms of their contribution to CO₂ exchange with the atmosphere at the global scale. This is a topic that is certainly relevant and timely, and recent previous work by some of the co-authors has highlighted that the gas exchange coefficient in such systems is likely to be much higher than assumed in previous efforts to upscale river CO₂ emissions. As such, it is good to see a manuscript that focuses specifically on mountainous streams. However, I feel there are many aspects that need to be clarified before this extrapolation can be considered for publication, and I have strong reservations on the interpretation of the stable isotope data. The idea underlying this ms is worth pursuing – but it needs a major overhaul to be convincing.

-L31: “groundwater deliveries of CO₂ from rock weathering”. Rock weathering consumes CO₂, it does not release CO₂. This is a confusing line of thought that should be clarified – the authors come back to this in the manuscript on L 218-220, where they mention “retrograde solubility” of carbonates, i.e. carbonates are weathered (consuming CO₂), and precipitate again in (warmer) streams, thereby releasing CO₂. There is no evidence provided for this in the ms – hence as a mechanism to contribute to CO₂ evasion it is too speculative.

-the term “geogenic” CO₂ first appears on L 215, but should be clearly defined – the authors use this term to refer to C derived from the dissolution of (fossil) carbonates, but in much of the literature (the Campeau et al. 2017 paper the authors refer to being an exception) the term is associated with CO₂ outgassed in geothermally active regions – which is a very different process.

-L33: the reported uncertainty on this estimate is extremely low – I do not see how this can be realistic. On L 229, the confidence interval of this estimate is reported to be 19-193 Tg C yr⁻¹. How can the confidence interval span an order of magnitude, with the resulting median value so close to the 95% CI? For the Swiss study sites, this looks very different (4.1 kg C m⁻² yr⁻¹, CI: 0.03 – 26.0 kg C m⁻² yr⁻¹) – L117.

-L34: “relatively contribution”: relatively low contribution?

-L36: “hitherto unrecognized contributors to global C fluxes”: that's not really correct, low order streams have been thought to be of particular importance in global CO₂ emissions from surface waters (e.g. Raymond et al. 2013). The merit of this manuscript is that there is an explicit focus on these systems, taking into account recent new insights on the gas exchange velocity in such systems.

-L51-52: “often lack significant vegetation cover and soil carbon stocks”: this is an overgeneralization. High-altitude regions can also develop highly organic soils and peatlands (e.g. paramo systems).

-L173, L177 and elsewhere: GloRIC database: should this be GloRICH database?

-L162 and further: The discussion of the d¹³C data is largely tucked away in the supplementary files. On the one hand, they would merit some more discussion in the main text, on the other hand their interpretation should be reconsidered. A few important points to consider:

* The reported offset between measured and predicted d¹³C-CO₂ values of 5.1 ± 2.2 ‰ (L162 and Figure S7) is odd. Looking at Figure S7A, these data appear to make perfect sense: CO₂ is depleted in ¹³C relative to the total DIC pool, as expected for equilibrium isotope fractionation. The offset appears to be in the order of 8 per mil, which is in line with expectations for a system where CO₂ is a minor

fraction of the total DIC. The data in Figure S7B, however do not make sense to me – I do not see how, with the given $\delta^{13}\text{C}$ -DIC data, one can get the pattern of expected $\delta^{13}\text{C}$ -CO₂ shown here. Isotope fractionation between bicarbonate and dissolved CO₂ is largest at low temperatures but does not exceed -10.8 ‰ – this is inconsistent with what I'm seeing when trying to pair the data in FigS7A and FigS7B.

More details should be provided on the above calculations; but irrespective of that, the fact that there is a difference of ~5 per mil between expected and measured $\delta^{13}\text{C}$ -CO₂ should not be equated to "isotopic enrichment due to high evasion rates" (caption of Figure S7, line 245-246): CO₂ evasion leads to a ^{13}C -enrichment in the remaining DIC pool, since CO₂ is depleted in ^{13}C relative to bicarbonate and carbonate. If there is an isotopic disequilibrium between CO₂ and other DIC species, this does not necessarily imply that gas evasion is the cause.

-page 7-8: The global extrapolation relies on the GloRICH database – which also formed the basis of Raymond et al. (2013) and Lauerwald et al. (2015). The authors should indicate to which extent new data were added in the meantime – or is it essentially the same global data, but with a new gas transfer velocity parameterization? Secondly, I was somewhat confused on whether the data used are only datasets with pCO₂ measured via direct methods? In that case, this is an important distinction with previous global extrapolations based on the GloRICH database.

-Miller-Tans plots: These are in fact, not really appropriate for the purpose they are used for here. I realize these have been used previously in a similar context (Campeau et al. 2017), and unfortunately there are numerous other studies in geochemistry where this approach is used inappropriately. It is obviously a conceptually and visually elegant approach, that often appears to lead to consistent results about source $\delta^{13}\text{C}$ values – however, one must keep in mind the underlying principles and context for which it applies. This approach was developed for simple systems where there is a certain background pool initially, and where a single source is added over time. This is entirely different from a complex range of streams where the background DIC concentration varies, and where there is a multitude of factors influencing both the concentrations and isotope composition of DIC: gas evasion, primary production, carbonate precipitation/dissolution, etc. One cannot expect to derive a meaningful estimate of the $\delta^{13}\text{C}$ of the 'source' of added DIC from such a dataset. Again – I realize this was done earlier in other studies but this is not a good justification – one should avoid making the same mistake again. In fact, when we look into Campeau et al. (2017), one will notice that they did not use the Keeling approach ($\delta^{13}\text{C}$ versus $1/\text{DIC}$) but the Miller-Tans approach – which will obviously result in some sort of correlation as one plots DIC versus DIC * $\delta^{13}\text{C}$ (i.e. A versus A*B) – but that does not make the result meaningful. Also, it is not clear why you applied this approach on both CO₂ and DIC, and what the underlying idea is.

-Materials and Methods, line 139 and further: Methods of analysing $\delta^{13}\text{C}$ -CO₂: (i) it's not indicated whether or not these samples were filtered, and whether they were preserved (e.g. with HgCl₂)? (ii) mention if (and how) the analysis corrected for isotope fractionation between gaseous CO₂ (as measured in the headspace) and dissolved CO₂. Irrespective of this, I do not see the point of this methodological approach: taking water samples and measuring $\delta^{13}\text{C}$ in CO₂ after several days/weeks in order to draw conclusions on whether or not CO₂ was in isotopic equilibrium in the streams. By the time you measure them, they will undoubtedly be in isotopic equilibrium – if the data suggest that they are not, there is something wrong with the methods. In short, based on the methodology as it is described the entire discussion on offsets between $\delta^{13}\text{C}$ -CO₂ and predicted $\delta^{13}\text{C}$ -CO₂ does not appear to be valid. The only way to measure $\delta^{13}\text{C}$ -CO₂ as it is 'in situ' would be to strip out the CO₂ immediately after sampling.

-Supplement, L151-153: what is meant with '... indicate a CO₂ source influenced by carbonate weathering (close to 0 per mil), in addition to a more isotopically depleted source'? First, carbonate weathering is a source adds DIC with a $\delta^{13}\text{C}$ intermediate between the source of CO₂ driving its dissolution and the carbonate-C (hence, not close to 0) and secondly, how does can this approach

indicate 2 sources ?

-Another potential source of groundwater pCO₂ data is Jurado et al. (2018), *Science of the Total Environment* 619-220: 1579-1588. I expect some of their sites to fit with your definition of 'mountainous'.

-Reference list needs some human interaction to clean it up, was generated automatically I assume. Eg. ref #14 has 'S.R. Geophysical' as last author, ref #23 has J.E. Richey as its only author, ref#30 has 'NMPEA Planetary' as last author, ref#41 is authored by K.M.E.A.P.S. Letters.. etc.

Reviewer #2 (Remarks to the Author):

The authors claim that CO₂ evasion fluxes from mountain streams equal or exceed those reported from tropical and boreal streams. They find that in the Swiss Alps groundwater contributes CO₂ from two sources: rock weathering and soil respiration. Extrapolating their results to the global scale, the authors estimate that 192 ± 2 Tg C yr⁻¹ is emitted from mountain streams, which would translate to a range of 10-30% of estimated global emissions from streams and rivers.

The paper represents a very timely and significant contribution to the ongoing scientific debate on the role of the land-ocean aquatic continuum in transferring terrestrial carbon to the atmosphere. Upscaling regional studies to the global scale is never easy and always requires an element of simplification and speculation. This study represents a huge effort by a competent group and I recommend publication if the main points of discussion (1-8 below) can be addressed or rebutted.

The main idea of the paper rests on the recent discovery that bubble entrainment governs gas exchange velocity in high-energy alpine streams in contrast to diffusive gas transfer in low-energy streams (Ref. 8 Ulseth et al. 2019, *Nature Geoscience*, 12, 259-263). As shown in this previous study, the gas-transfer rate k_{600} in mountain rivers can be estimated from a scaling relation with energy dissipation. The authors use hydraulic scaling relations for mountain rivers of up to a discharge of 2.26 m³ s⁻² and a global river network and discharge data to calculate k_{600} for almost 2 million stream segments in mountain areas. This part of the study opens an exciting perspective to gas transfer in mountain streams.

In order to predict CO₂ transfer, the authors document the procedure how to calculate CO₂ fluxes from dissolved CO₂ concentrations in the rivers. They use standard equations for estimating diffusive fluxes of CO₂ between water and air (SI, pages 3-4). At present, this seems to be the best available process, but ironically, the authors have shown in their Ref 8, that these equations break down for steep mountain rivers because air bubbles dominate gas transfer. As in the ocean, the bubbling regime will induce supersaturation via excess air. It is unclear how to model these non-equilibrium processes exactly. The authors should acknowledge the limitation of modeling bubble entrainment as if it were a standard process of gas diffusion.

In a next step, the study presents a linear statistical model for predicting CO₂ concentrations based on global datasets for elevation, discharge and soil organic carbon (methods line 62). This model is the weak part of workflow for the upscaling process. There are several reasons:

1. Sampling bias. Comparing the global map of input data for the statistical model in Figure S3A with the main result in Figure 2D, it becomes evident that large parts of the world's mountains are not covered: The Andes, most of the African Highlands, the volcanic terrain of the Pacific Rim, South- and South-East Asia etc.

2. Range of altitude data: The altitude distribution of samples looks fine for the range of 400 to 3000 m (Fig. S3B). Therefore, the model does not cover high mountain areas like the Tibetan plateau

(Figure S10). Altitudes significantly above 3500 m should therefore be excluded in the analysis.

3. Range of discharge samples: The discharge data look strange. In Fig S3-C, they range from about 1 m³ s⁻¹ (ln discharge = 0) to less than a few milliliters per second (ln discharge = -12). This would mean that a significant part of the CO₂ data are from stream sections that are not covered by global data sets. The Swiss sample reanges from 0.02 – 2 m³ s⁻¹ which would translate to a lower ln limit of about -4. Is Fig. S3-C correct?

4. Range of soil data. Most of the soil organic carbon (SOC) data are centered within the high range (10-40%, ln[SOC] = 4.5-6 g kg⁻¹). This is problematic because in general SOC decreases with altitude and reflects the type of vegetation cover.

5. Model performance: The model narrows the two orders of magnitude in observed CO₂ concentration data (3 - 400 micromolar) down to a factor of 5 in the predicted range (Figure S3E). This raises questions, whether the model approach is really useful: Global estimates would probably not change significantly if just the median value and percentiles for the CO₂ concentration were used in the global calculations.

The authors should critically review their database and expand it or discuss the limitations and uncertainties of the model more explicitly. One obvious way to expand the database is the use of CO₂ values obtained from alkalinity and pH. Although the quality of wet-chemistry data at low pH, low alkalinity and high DOC is questionable, the large remaining set of data will significantly improve the statistical power of the model outlined in Fig S3. Two additional governing factors require attention:

6. Geology: An extended database of CO₂ in rivers would also allow the correlation with geology. Weathering of carbonate rock is a clear feature of the Swiss data (Fig S6). There is a need for more coverage of terrains with igneous rocks or basalt. (See the GLiM database by Hartmann and Moorsdorf, 2012 G3 13, Q12004.)

7. Seasonality: Outside the tropics, seasonality in river flow increases dramatically with altitude. Freezing temperatures and snow cover will reduce stream flow in the cold season, so that typical field observations only cover half of the year.

The different weak spots in the model for predicting CO₂ concentrations in mountain rivers lead to the key question: Are the high CO₂ emission rates predicted in this paper for the world's mountain streams plausible?

8. Groundwater inflow: As a partial answer, the authors perform a validation exercise for the 4000 Swiss streams. They calculate the CO₂ concentration in the groundwater needed to support the evasion rates and compare those with "literature data" (Figure S5). Table S2 reveals that these literature data are all personal communications without any additional information. For a proper validation these values should be cleaned up (nobody measures CO₂ in water to 5 digits precision), the data need to be georeferenced and a comparison between the CO₂ mass balance and the measurements should be given for the different sites.

9. For the global estimate of a CO₂ emission by mountain streams of almost 200 Tg per year or 10-30% of the global emission rates, 650 – 1800 Tg per year) this study should show more convincingly where the carbon comes from. High mountain terrains with their high k₆₀₀ values exhibit low primary production and often short seasons for soil respiration. The authors should address this question more clearly.

Minor comments:

line 32 "groundwater CO₂ deliveries from rock weathering and soil respiration" - this is a bit a side

track, because it has been known since the work of Garrels, Berner and others in the 1980ies that rock weathering transfers atmospheric CO₂ to the hydrosphere. A detailed discussion of weathering versus soil respiration would call for an expanded model with geological information (see remark 6) line 33: Not clear what the 1% uncertainty refers to (192 +- 1.9 Tg C yr⁻¹). In the light of the many model limitations outlined above such a high precision seems questionable. The 5 and 95% error bands in Figures 2 E – H seem to tell a different story.

lines 137 138, realistic precision for CO₂ values needed (3 significant digits is quite demanding). This paragraph refers to data from different parts of the world, but neither Figure S5 nor Table S2 provide georeferenced information. It is strange that the data calculated for 4000 Swiss river segments are compare to a random sample of data from the Chez Republic and Laos.

line 146 “we were not able to include alkalinity as a potential sink for CO₂ in the mass balance”. Why not? By the way, the pool of carbonate alkalinity could also act as a source if stripped with air bubbles.

line 273 – formatting ref 5

line 279 ref 8 needs updated page numbers

line 311 clean up citation Butmann et al.

line 318 update ref 23.

line 368 legend Figure 1 should mention that these are modelled distributions. Not clear what “multiple” stable isotopic analyses means. There is only one isotope measured.

Figure 1 and other Figures in the supporting information: The exponents d-1 in Figure A and m-2 yr-1 in Figure C are not formatted correctly.

Figure S5 There are no literature values in this supplement – only personal information.

Reviewer #3 (Remarks to the Author):

Overview – Horgby et al present a compelling analysis that improves upon many of the recent estimates of stream and river CO₂ emissions. In particular, their work focuses on high elevation systems that are based upon recent findings from a 2 year effort in the Swiss Alps. Their works supports the hypothesis that groundwater and soil respiration contribute significant inorganic carbon to small mountainous streams, that that the physical environment of high slope and turbulent conditions creates conditions where evasion is high resulting in low measured CO₂ concentrations. Using a simple mass balance approach, the authors support the potential source of groundwater carbon dioxide within these systems. This manuscript is well done, and the analyses are complete. However, it is a shame that much of the analysis is lost within the supporting information and may never get highlighted as a result of the short format of Nature Communications.

Major Comments:

There are two aspects of this work that I believe warrant more explicit discussion for this to be published, and I am sure that these can be handled well by the authors. Within the manuscript as written, there is very limited discussion on the temporal nature of CO₂ concentrations in streams. This is very important when attempting to scale to annual estimates. This reviewer acknowledges that datasets are not yet available at a scale to properly constrain temporal dynamics globally, there should be more explicit discussion – perhaps based on the findings from the continuous monitoring within the Alps, how deviation in high elevation concentrations and subsurface CO₂ production may influence these global estimates. Along these lines, is it possible to provide a context for what 2000m means in the tropics vs. northern latitudes? If a system freezes – what is the impact on the annual emissions. If this was discussed explicitly, this reviewer did not see it. Do the authors assume no emissions for a portion of the year when frozen? Does precipitation drive emissions at all? It was surprising that this did not factor into the simple linear model to predict CO₂ concentrations on its

own.

Given the very large potential range in the predictors used to model emissions, it is surprising that the estimated error derived from the Monte Carlo for flux is only 1% at 1.9 Tg-C. In fact this level of precision is somewhat suspect. The authors could provide some additional clarity on how this was developed and reduced and what that might mean for interpretations.

The only other aspect that should be addressed is the potential impact of the available datasets on stream width and hence surface area. This reviewer agrees with the authors that this work could be considered conservative. In fact the spatial datasets used for the global analysis appears to only work at a resolution of 10m or greater? It is recommended that the authors discuss the potential loss of streams below these thresholds if possible in more detail. The authors could utilize the cited paper Allen et al. 2018 – Nature Comm. (<https://doi.org/10.1038/s41467-018-02991-w>) This citation details a potential model for capturing very small streams in an area calculation.

Minor comments:

NO REFERENCE 5?

34 – relatively...

35 – hitherto... awkward

69 – used per/mil not percent

102-106 – can the authors bring in more description of how the error of using the relatively weak linear model propagates into the estimates of CO₂ concentration here?

122 – is geopredictors a real term?

226 – change culminating to summing...

Methods –

When converting air temperature to water temperature, was there any analysis that suggest these systems that are governed by turbulence across adhere to the cited equation? Also - was this component of the analysis included within the Monte Carlo assessment?

112 – converted...

148- can you provide a figure for how the miller tans approach separated the potential sources? This can be added to the supporting information with. It would appear that there is a 10 per mil range in the swiss alps dataset, are there additional datasets that can contribute here?

Reviewers' comments:

Reviewer #1 (Remarks to the Author):

Review of Horgby et al: Unexpected large evasion fluxes of carbon dioxide from turbulent streams draining the world's mountains.

The manuscript by Horgby et al. assesses the importance of mountainous streams and rivers in terms of their contribution to CO₂ exchange with the atmosphere at the global scale. This is a topic that is certainly relevant and timely, and recent previous work by some of the co-authors has highlighted that the gas exchange coefficient in such systems is likely to be much higher than assumed in previous efforts to upscale river CO₂ emissions. As such, it is good to see a manuscript that focuses specifically on mountainous streams. However, I feel there are many aspects that need to be clarified before this extrapolation can be considered for publication, and I have strong reservations on the interpretation of the stable isotope data. The idea underlying this ms is worth pursuing – but it needs a major overhaul to be convincing.

AUTHORS: We are grateful for the overall positive impression that this reviewer has on our study. We also share with her/him that the manuscript does benefit from a major “overhaul” to better highlight the novelty of our work. Novelty is along three major lines at least: (i) for the first time, global CO₂ fluxes from mountain streams are being computed; (ii) rather than using aggregation approaches as typically done in previous studies (e.g. Raymond et al. 2013; Lauerwald et al. 2015) we do compute CO₂ fluxes for each stream individually, which involves a novel approach to uncertainty estimation; (iii) we combine our flux measurements with data on stable isotopes and a “back-of-the-envelope” mass balance calculation for CO₂ contributions from groundwater. We are convinced that this study will be well received by the scientific community. At the same time we are aware of the numerous potential caveats inherent to any extrapolation effort; we have now better discussed them in the revised manuscript by devoting an entire section to uncertainties and limitations of extrapolating CO₂ fluxes from mountain streams.

Essentially, this reviewer with her/his comments (L31, L215 and L162) and more specifically on the use of Miller-Tans plots questions our use of stable isotopes and mixing analyses to infer potential sources of CO₂. We will here give a brief but generic response to this overall criticism to then come to each of her/his comments in some more detail.

While we do not fully agree with some of the criticisms of this reviewer related to stable isotope analyses, which is in the very nature of the scientific debate, we do comply with her/him and abstained from showing the results from the Miller-Tans plots. Instead, we do show the results from the DIC Keeling plots as this is a very widely used and recognized technique to identify sources of DIC in the ocean, lakes, streams and rivers (e.g., Karlsson et al. 2007 Limnology and Oceanography; Karlsson et al. 2008 Limnology and Oceanography; Mortazavi and Chanton 2004 Limnology and Oceanography; Drake et al. 2018 JGR Biogeosciences; Horgby et al. 2019 JGR Biogeosciences; Campeau et al. 2018 JGR Biogeosciences; Campeau et al. 2018 Scientific Reports). It may be worth mentioning that the findings from the CO₂ Miller-Tans plots as shown in the original submission agree very well with the findings from the DIC Keeling plots. We do hope that our decision to focus on the latter, also its wide acceptance by the scientific community, relieves some of the reservation that this reviewer had.

Next, this reviewer also had concerns as for our proposition that “retrograde solubility” could play a role in the CO₂ dynamics in certain mountain streams. We gratefully acknowledge her/his point, which we discuss in some detail below.

-L31: “groundwater deliveries of CO₂ from rock weathering”. Rock weathering consumes CO₂, it does not release CO₂. This is a confusing line of thought that should be clarified – the authors come back to this in the manuscript on L 218-220, where they mention “retrograde solubility” of carbonates, i.e. carbonates are weathered (consuming CO₂), and precipitate again in (warmer) streams, thereby releasing CO₂. There is no evidence provided for this in the ms – hence as a mechanism to contribute to CO₂ evasion it is too speculative.
-the term “geogenic” CO₂ first appears on L 215, but should be clearly defined – the authors use this term to refer to C derived from the dissolution of (fossil) carbonates, but in much of the literature (the Campeau et al.2017 paper the authors refer to being an exception) the term is associated with CO₂ outgassed in geothermally active regions – which is a very different process.

AUTHORS: This is an important comment for which we are grateful indeed. Addressing this more properly will improve our manuscript. First, we acknowledge that we have not well enough explained the terminology. For instance, biogenic and geogenic are two terms that are very widely used in the relevant literature (Johnson, Billett, Wallin, Duvert etc) to refer to CO₂ originating from the decomposition of organic carbon with the ultimate production of CO₂ and to the dissolution of carbonates (mostly), respectively. We are using lithogenic in our study, rather than geogenic, and where suitable, we simply refer to “soil respiration”.

As a response to the comment on the “retrograde solubility” and carbonate weathering consuming CO₂ — also linking to the L162 comment (below) of this reviewer — we agree that the process has been poorly described and at the end remains speculative. We may emphasize that the prime goal of our study is the upscaling of CO₂ fluxes from mountain streams. Given the high areal fluxes, it is intuitive to ask where all that CO₂ comes from. Here we present an ensemble of groundwater mass balance calculations over more than 3000 streams and our stable isotope analyses. While the patterns are clear, underlying processes remain more speculative. We do better acknowledge this in the revised version now.

We believe that our reasoning behind the “retrograde solubility” is correct and our line of argumentation reads as follows:

- 1. Rock weathering or the dissolution of minerals (e.g., carbonates) fixes CO_{2atm} during the alteration process as the carbonic acid in water. This acid reacts both with silicates and carbonates, adding dissolved inorganic carbon (DIC) into solution. For carbonate rock weathering, every mole of carbonic acid can react one mole of carbonate. The overall reaction thus delivers 2 moles of DIC to the solution. ($H_2O + CO_2 + CaCO_3 = 2HCO_3^- + Ca^{2+}$)*
- 2. Exfiltration of such a solution in high-altitude catchments, from the soil horizon and/or groundwater, will often allow the aqueous solution to warm up in summer (nota bene, water can warm up remarkably as it flows over exposed rock). Decreasing solubility of CO₂ with increasing temperature (according to Henry’s law) causes this “retrograde solubility” of carbonates, where the loss of CO₂ from the water (and increasing pH) is compensated by the precipitation of calcium carbonate.*
- 3. Now the proof of a “geogenic” (or lithogenic) contribution (defined as carbonate rock-derived DIC and ultimately of CO₂) of evasive CO₂ would come from the isotopic composition and concentration of the DIC. Considering the equation above, and using a CO_{2atm} δ¹³C value of -8.6 ‰, and assuming the dissolution of CO₂ as well as dissolution of marine carbonate (typically -2 to +2 ‰ in δ¹³C) to form DIC as equilibrium processes, this would potentially give δ¹³C(DIC as HCO₃ ion) values of about +1.5 ‰ at 5 °C and +0.9 ‰ at 10 °C (both 0.4 ‰ lower for the carbonate ion...but at our natural pH range this ion is only about 10% of the DIC; same for CO_{2aq} as part of the DIC – this would have δ¹³C of -9.8 ‰ as CO_{2aq} dissolved in water at 5 and 10 °C but represents also less than 10 % unless our pH is well below 7). In*

short, CO_{2atm} dissolved in water only as DIC would have a $\delta^{13}C$ -DIC of somewhere between 0.4 and 1.4 ‰ (pH between 6.5 and 8, respectively) if only CO_2 atm is involved. With a “geogenic” contribution of marine carbonate dissolution this atmospheric CO_2 obtained DIC would mix in a 1:1 molar ratio with a DIC derived from carbonates with $\delta^{13}C$ of -5.3 ‰ for -2 ‰ carbonate and of -1.3 ‰ for +2 ‰ carbonate. Hence, the range of $\delta^{13}C$ of the mixture would now be somewhere between -2.5 and 0 ‰ but with twice the amount of DIC compared to the solubility of an atmospheric CO_2 of about 400 ppm. As such, the indication of a geogenic carbonate component to our CO_2 would be difficult to distinguish from a purely isotopic point of view, but if DIC values of $\delta^{13}C$ are close to -3 or higher ‰ at concentrations higher than CO_2 concentrations of our waters at 5 to 10 °C, for example (based on atmospheric CO_2 contribution only), this would support a lithogenic component to DIC.

4. If soil organic matter decomposition (forming CO_2) contributes to the DIC pool, rather than, for example CO_{2atm} , this CO_2 would have $\delta^{13}C$ values of DIC (equilibrium dissolution of this respired CO_2) of about -26 ‰ for CO_{2aq} DIC and -15 ‰ for CO_2 dissolved as HCO_3^- , hence for pH of about 6.5 a total of about -16 ‰ and -14.6 ‰ at a pH of 8, but if mixed with a ratio of 1:1 with “lithogenic” derived CO_2 these mixed solution DIC values would be between about -10.7 and -8 ‰, all at 5 °C.

5. Now, if we look at our measured DIC concentrations and $\delta^{13}C$ values of DIC, which are between -11 and -3 ‰, and apply a linear mixing line (Keeling plot) in a $1/conc.$ vs $d^{13}C$ plot, this would support lithogenic and organic sources of DIC that will, upon warming the waters gently release CO_2 to the atmosphere as they are generally oversaturated, due to the “retrograde solubility” of carbonates. Thus, this supports our statement in the manuscript; however, we agree that we cannot provide the ultimate evidence for this.

As a reaction to this point, we have now toned down the idea of “retrograde solubility”, added a reference (Drysdale et al. 2003, Hydrological Processes), and we have included a more complete explanation of “retrograde solubility” in the SI.

Perhaps more important, we report the alkalinity of the Swiss streams, but also of the streams contained with the GLORICH data base, to make the point that indeed many of these streams are candidates for the DIC from lithogenic sources driving the CO_2 supersaturation — in analogy to the recent Marce et al. paper in Nature Geoscience on lakes.

-L33: the reported uncertainty on this estimate is extremely low – I do not see how this can be realistic. On L 229, the confidence interval of this estimate is reported to be 19-193 Tg C yr⁻¹. How can the confidence interval span an order of magnitude, with the resulting median value so close to the 95% CI? For the Swiss study sites, this looks very different (4.1 kg C m⁻² y⁻¹, CI: 0.03 – 26.0 kg C m⁻² yr⁻¹) – L117.

AUTHORS: Thanks for carefully reading. This is of course a most embarrassing typo – the “zero” was missing and it should have read 190 – 193 Tg C yr⁻¹. We have updated the flux estimates now, with the changes made on the CO_2 model incorporated (see further below) as well as including an alternative approach to simulate the uncertainty (see responses to reviewer #2). Furthermore, please note that the fluxes from the Swiss sites the reviewer is referring to are areal fluxes (kg C m⁻² y⁻¹) — not to the extrapolation to the Swiss Alps! Please note that the uncertainty associated with the areal fluxes differs from the uncertainty associated with the global estimate (Tg C yr⁻¹). The revised version also contains the total flux for the Swiss Alps.

We would like to kindly underscore the fact that that we have now dedicated a substantial new section to the calculation and discussion of the various uncertainties in the revised manuscript and in the method section (please, see below).

-L34: “relatively contribution”: relatively low contribution ?

AUTHORS: Thanks. Changed accordingly.

-L36: “hitherto unrecognized contributors to global C fluxes”: that’s not really correct, low order streams have been thought to be of particular importance in global CO₂ emissions from surface waters (e.g. Raymond et al. 2013). The merit of this manuscript is that there is an explicit focus on these systems, taking in to account recent new insights on the gas exchange velocity in such systems.

AUTHORS: Thanks for this comment. Pete Raymond’s paper makes the case that k_{600} is higher in mountain streams, and at the same time, it emphasizes the role of headwaters for global CO₂ fluxes. However, not all headwaters are mountain streams. Here we underscore the role of mountain (!) streams for global CO₂ fluxes. We have changed that sentence accordingly.

Furthermore, we also want to stress the fact that Raymond et al. (2013) have scaled k_{600} values and estimates of stream surface areas, assuming all stream orders within a region to show the same pCO₂. These regionalizations are not suitable to distinguish low land from mountain streams in a region and they used averages instead, which might additionally blur the very specific attributes of mountain streams.

-L51-52: “often lack significant vegetation cover and soil carbon stocks”: this is an overgeneralization. High-altitude regions can also develop highly organic soils and peatlands (e.g. paramo systems).

AUTHORS: This is an important point for which are grateful. We have changed this sentence accordingly, which also relates to the SOC comment from reviewer 2 (please, see below).

-L173, L177 and elsewhere: GloRiC database: should this be GloRICH database ?

AUTHORS: The Global River Classification (GloRiC) database is different from the Global River Chemistry database (GLORICH). The GloRiC database (<https://www.hydrosheds.org/page/gloric>) provides data on river hydrology, physio-climatology, geomorphology, for instance. One of our co-authors, Bernhard Lehner, is the PI of the GloRiC database. The GLORICH database (<https://www.geo.uni-hamburg.de/en/geologie/forschung/geochemie/glorich.html>) provides relevant data on river hydrochemistry and has served as the basis for most exercises towards global CO₂ evasion estimates — including the ones by Ronny Lauerwald, also a co-author on our manuscript. Please, see our comments below why we decided not to use the GloRiC data.

We have clarified that sentence in the manuscript.

-L162 and further: The discussion of the d13C data is largely tucked away in the supplementary files. One the one hand, they would merit some more discussion in the main text, on the other hand their interpretation should be reconsidered. A few important points to consider:

* The reported offset between measured and predicted d13C-CO₂ values of 5.1 ± 2.2 ‰ (L162 and Figure S7) is odd. Looking at Figure S7A, these data appear to make perfect sense: CO₂ is depleted in ¹³C relative to the total DIC pool, as expected for equilibrium isotope fractionation. The offset appears to be in the order of 8 per mil, which is in line with expectations for a system where CO₂ is a minor fraction of the total DIC.

The data in Figure S7B, however do not make sense to me – I do not see how, with the given

d13C-DIC data, one can get a the pattern of expected d13C-CO2 shown here. Isotope fractionation between bicarbonate and dissolved CO2 is largest at low temperatures but does not exceed -10.8 ‰ – this is inconsistent with what I'm seeing when trying to pair the data in FigS7A and FigS7B.

More details should be provided on the above calculations; but irrespective of that, the fact that there is a difference of ~5 per mil between expected and measured d13C-CO2 should not be equated to “isotopic enrichment due to high evasion rates” (caption of Figure S7, line 245-246): CO2 evasion leads to a 13C-enrichment in the remaining DIC pool, since CO2 is depleted in 13C relative to bicarbonate and carbonate. If there is an isotopic disequilibrium between CO2 and other DIC species, this does not necessarily imply that gas evasion is the cause.

AUTHORS: We are grateful for this comment and would like to refer the reviewer also to our discussion above. Given the reservations of this reviewer, we have decided to remove the analyses based on $\delta^{13}\text{C-CO}_2$ from our study. Instead, we now focus on the DIC Keeling plots.

-page 7-8: The global extrapolation relies on the GloRICH database – which also formed the basis of Raymond et al. (2013) and Lauerwald et al. (2015). The authors should indicate to which extent new data were added in the meantime – or is it essentially the same global data, but with a new gas transfer velocity parameterization? Secondly, I was somewhat confused on whether the data used are only datasets with pCO2 measured via direct methods? In that case, this is an important distinction with previous global extrapolations based on the GloRICH database.

AUTHORS: We understand that this comment emanates from a misunderstanding of the GLORICH versus GloRiC databases as briefly discussed above. We did not use CO2 data calculated from the GLORICH database as done by Ronny Lauerwald and Pete Raymond, for instance. We strictly used a novel data set of directly measured CO2 data from mountain streams. Reason for this is that direct CO2 measurements from many various settings is less prone to error than CO2 data calculated from alkalinity and pH, as is now widely accepted (e.g., Abril et al. 2015; also Raymond et al. 2013 in SI) and as we alluded to in our manuscript. For instance, Raymond et al. (2013) found calculated median CO2 concentrations (from aggregated data over SE Asia) of 100,000 μatm — that were biasing interpolations. For these same reasons, CO2 data from the GLORICH database are often aggregated over very large regions and the median value is equally used for all streams and rivers within that region (e.g., Raymond et al., 2013). This approach can add uncertainty to the upscaling (please, see below where we discuss our uncertainty calculations) something that is especially important considering our mountain streams due to the overall low pCO2 values. A bias in the estimate could therefore have large consequences, where for instance 200 μatm could lead to errors of up to 50 % of the estimated pCO2.

While our approach using directly measured CO2 data gives more reliable data — notably for systems with very low CO2 concentrations, it does come at the cost of a lower sample size — we are well aware of this trade-off.

Alternatively, it could well be that the reviewer also referred to the GloRiC database with its HYDROSHEDS data and asks which new data were added to this database in the meantime (we are mostly sorry, but her/his query seems unclear to us). In that case, we are delighted that we used the recently published GloRiC database. These data allow a better quantification of the streams above the 60° northern latitude. The lack of these data was identified as a weakness by Pete Raymond in his Nature paper, for instance. We do acknowledge though that this still comes at the costs of a reduced spatial resolution compared to data below the 60° northern latitude. Nevertheless, this is one of the many points where we are convinced that our extrapolation efforts provide a better appreciated of global CO2 fluxes, including the associated error, compared to previously published estimates.

-Miller-Tans plots: These are in fact, not really appropriate for the purpose they are used for here. I realize these have been used previously in a similar context (Campeau et al. 2017), and unfortunately there are numerous other studies in geochemistry where this approach is used inappropriately. It is obviously a conceptually and visually elegant approach, that often appears to lead to consistent results about source $\delta^{13}\text{C}$ values – however, one must keep in mind the underlying principles and context for which it applies. This approach was developed for simple systems where there is a certain background pool initially, and where a single source is added over time. This is entirely different from a complex range of streams where the background DIC concentration varies, and where there is a multitude of factors influencing both the concentrations and isotope composition of DIC: gas evasion, primary production, carbonate precipitation/dissolution, etc. One cannot expect to derive a meaningful estimate of the $\delta^{13}\text{C}$ of the ‘source’ of added DIC from such a dataset. Again – I realize this was done earlier in other studies but this is not a good justification – one should avoid making the same mistake again. In fact, when we look into Campeau et al. (2017), one will notice that they did not use the Keeling approach ($\delta^{13}\text{C}$ versus $1/\text{DIC}$) but the Miller-Tans approach – which will obviously result in some sort of correlation as one plots DIC versus $\text{DIC} * \delta^{13}\text{C}$ (i.e. A versus $A*B$) – but that does not make the result meaningful. Also, it is not clear why you applied this approach on both CO_2 and DIC, and what the underlying idea is.

AUTHORS: We are grateful for this comment, despite the fact that we do not necessarily agree with some of the arguments brought forward (e.g. the apparent spurious correlation when slopes matter). Nevertheless, as made clear above, we abstain from showing the Miller-Tans approach and focus on the Keeling approach, well rooted in aquatic and terrestrial (bio)geochemistry (please see above).

-Materials and Methods, line 139 and further: Methods of analysing $\delta^{13}\text{C}$ - CO_2 : (i) it’s not indicated whether or not these samples were filtered, and whether they were preserved (e.g. with HgCl_2) ? (ii) mention if (and how) the analysis corrected for isotope fractionation between gaseous CO_2 (as measured in the headspace) and dissolved CO_2 . Irrespective of this, I do not see the point of this methodological approach: taking water samples and measuring $\delta^{13}\text{C}$ in CO_2 after several days/weeks in order to draw conclusions on whether or not CO_2 was in isotopic equilibrium in the streams. By the time you measure them, they will undoubtedly be in isotopic equilibrium – if the data suggest that they are not, there is something wrong with the methods. In short, based on the methodology as it is described the entire discussion on offsets between $\delta^{13}\text{C}$ - CO_2 and predicted $\delta^{13}\text{C}$ - CO_2 does not appear to be valid. The only way to measure $\delta^{13}\text{C}$ - CO_2 as it is ‘in situ’ would be to strip out the CO_2 immediately after sampling.

AUTHORS: This comment relates to the comment above (L162). We have analyzed our samples within 24 h after sampling. Given the potential caveats with this, in combination with the reservations this reviewer has for the Miller-Tans analyses, we have now removed the latter from the manuscript. With this change, a lengthy response to this comment becomes obsolete.

-Supplement, L151-153: what is meant with ‘... indicate a CO_2 source influenced by carbonate weathering (close to 0 per mil), in addition to a more isotopically depleted source’ ? First, carbonate weathering is a source adds DIC with a $\delta^{13}\text{C}$ intermediate between the source of CO_2 driving its dissolution and the carbonate-C (hence, not close to 0) and secondly, how does can this approach indicate 2 sources?

AUTHORS: As above, given that we have now removed the Miller-Tans analyses, we have removed that part from the revision.

-Another potential source of groundwater pCO₂ data is Jurado et al. (2018), Science of the Total Environment 619-220: 1579-1588. I expect some of their sites to fit with your definition of 'mountainous'.

AUTHORS: We are very grateful for this suggestion, particularly data on shallow groundwater CO₂ concentration are very scarce. That is why we had included in our original submission even data from Laos, Czech Republic and low-land Switzerland. Following a comment from reviewer 2, we have now excluded these data from our compilation. For the same reason, we were not able to include the data from the Jurado et al. (2018) study in our comparison, since none of the locations were in areas classified as "mountains" according to the definition from Meybeck et al. (2001). However, we added two data points from one of our other study catchments (catchment B; Figure S1) measured this month by one of our colleagues. Moreover, we added a couple of sentences discussing that groundwater pCO₂ might be much greater than the median value that we have estimated as required to sustain estimated evasion fluxes.

-Reference list needs some human interaction to clean it up, was generated automatically I assume. Eg. ref #14 has 'S.R. Geophysical' as last author, ref #23 has J.E. Richey as its only author, ref#30 has 'NMPEA Planetary' as last author, ref#41 is authored by K.M.E.A.P.S. Letters.. etc.

AUTHORS: We are very sorry for this. Thanks for pointing out. All fixed now.

Reviewer #2 (Remarks to the Author):

The authors claim that CO₂ evasion fluxes from mountain streams equal or exceed those reported from tropical and boreal streams. They find that in the Swiss Alps groundwater contributes CO₂ from two sources: rock weathering and soil respiration. Extrapolating their results to the global scale, the authors estimate that 192 +- 2 Tg C yr⁻¹ is emitted from mountain streams, which would translate to a range of 10-30% of estimated global emissions from streams and rivers.

The paper represents a very timely and significant contribution to the ongoing scientific debate on the role of the land-ocean aquatic continuum in transferring terrestrial carbon to the atmosphere. Upscaling regional studies to the global scale is never easy and always requires an element of simplification and speculation. This study represents a huge effort by a competent group and I recommend publication if the main points of discussion (1-8 below) can be addressed or rebutted.

AUTHORS: We are very grateful to this reviewer as she/he unveiled some of the problems that are inherent to the extrapolation of biogeochemical fluxes to the regional and even global scale. It is evident that we cannot address them all to the full contention of the various colleagues in the field – neither to our own, of course. Our revised manuscript now includes a better assessment and critical discussion of our approaches. Nevertheless, this reviewer recognized the effort that we made to advance the field of upscaling CO₂ fluxes in mountain streams. Below we address each of the points that indeed we consider critical as well.

The main idea of the paper rests on the recent discovery that bubble entrainment governs gas exchange velocity in high-energy alpine streams in contrast to diffusive gas transfer in low-energy streams (Ref. 8 Ulseth et al. 2019, Nature Geoscience, 12, 259-263). As shown in this previous study, the gas-transfer rate k₆₀₀ in mountain rivers can be estimated from a scaling relation with energy dissipation. The authors use hydraulic scaling relations for mountain rivers of up to a discharge of 2.26 m³ s⁻² and a global river network and discharge data to calculate k₆₀₀ for almost 2 million stream segments in mountain areas. This part of the study

opens an exciting perspective to gas transfer in mountain streams.

In order to predict CO₂ transfer, the authors document the procedure how to calculate CO₂ fluxes from dissolved CO₂ concentrations in the rivers. They use standard equations for estimating diffusive fluxes of CO₂ between water and air (SI, pages 3-4). At present, this seems to be the best available process, but ironically, the authors have shown in their Ref 8, that these equations break down for steep mountain rivers because air bubbles dominate gas transfer. As in the ocean, the bubbling regime will induce supersaturation via excess air. It is unclear how to model these non-equilibrium processes exactly. The authors should acknowledge the limitation of modeling bubble entrainment as if it were a standard process of gas diffusion.

AUTHORS: We are grateful for this comment. Indeed, it is the scaling relationships that we published in Nature Geoscience that allow the community to properly predict k_{600} for steep-slope mountain streams. We want to emphasize that beside this, our upscaling also includes the following novel aspects: (i) We do not work on an aggregation basis as many others do (see Raymond et al., 2013 for instance). Rather we compute k_{600} and CO₂ for each stream individually, and hence the CO₂ flux. This is less prone to error as aggregating over very large regions and working with median values from there — this has recently been highlighted by Rocher-Ros et al. in Limnology and Oceanography Letters (2019). (ii) Based on this, we use a solid uncertainty estimation that we have now better developed and discussed than in the original submission. (iii) We do use direct CO₂ measurements rather than values calculated from pH and alkalinity – as discussed in some more detail above.

This reviewer understood that we used the “standard equations for estimating diffusive fluxes of CO₂” despite the fact that we based our premise on the Ulseth et al. (Nature Geoscience) k_{600} scaling relationships for white waters.

This is not true. Of course, we did use the Ulseth k_{600} scaling relationships. However, we do understand why this reviewer got misled in her/his understanding and we truly have to apologize for this. Reason for this is as follows:

In the Method section (original submission, lines 42 to 49) we show that we used the Ulseth et al. (Nature Geoscience) k_{600} scaling relationships as on energy dissipation (eD). Line 69 (same version) refers to the SI where we show how we converted k_{600} to k_{CO_2} . We do acknowledge that this way of unfolding the workflow is problematic, indeed, as it does not show the full string (one would have to jump from the Method section to the SI). Furthermore and even more important, in the SI, we depicted equation S6 (line 103, original version) in its classical way, without making the point that we solved it of course for k_{CO_2} , not for k_{600} .

On behalf of my colleagues and myself as the senior author on this manuscript, I have to sincerely apologize again for this awkward way to depict our workflow. The bottom line is that the CO₂ fluxes were not computed from the classical equation for diffusive gas exchange; they were calculated as from the Ulseth et al. approach.

In a next step, the study presents a linear statistical model for predicting CO₂ concentrations based on global datasets for elevation, discharge and soil organic carbon (methods line 62). This model is the weak part of workflow for the upscaling process. There are several reasons:

AUTHORS: We are most grateful for this series of comments to which we will now reply one by one. Before doing so, we would like to refrain that we prefer working with such a statistical model based on CO₂ directly measured in mountain streams and various “geopredictors”. For reasons discussed above, decided to abstain from using calculated CO₂ values (from the GLORICH database).

1. Sampling bias. Comparing the global map of input data for the statistical model in Figure S3A with the main result in Figure 2D, it becomes evident that large parts of the world's mountains are not covered: The Andes, most of the African Highlands, the volcanic terrain of the Pacific Rim, South- and South-East Asia etc.

AUTHORS: We do absolutely agree with the reviewer on this point and truly regret the poor data availability. As suggested by this same reviewer (see below), we could expand our database using data from GLORICH, for instance. Fact is, that the GLORICH database covers more or less the same regions as our data. To illustrate this, we have produced the following map showing the overlap between our data and the GLORICH data. To do so, we have extracted all data from the GLORICH database that match our requirements as “small mountain streams”, (i.e., based on the mountain classification from Meybeck et al. (2001) and with discharge lower than $2.26 \text{ m}^3 \text{ s}^{-1}$ (from Dallaire et al., 2018)). We do acknowledge that the GLORICH database may include more “shots” per region. However, owing to the very large uncertainty associated with the CO₂ estimates from alkalinity and pH, colleagues (e.g., Pete Raymond, Ronny Lauerwald) often use the median per region or per sampling location thereby considerably boiling down the sample size.

In our revised version, we make the point that more direct measurements of CO₂ are urgently required to better cover the various mountain ranges as highlighted by this reviewer. With this manuscript, we do show that mountain streams matter for global CO₂ fluxes and our hope is that this will spark further research in that direction.

2. Range of altitude data: The altitude distribution of samples looks fine for the range of 400 to 3000 m (Fig. S3B). Therefore, the model does not cover high mountain areas like the Tibetan plateau (Figure S10). Altitudes significantly above 3500 m should therefore be excluded in the analysis.

AUTHORS: It is clear that this criticism, very legitimate without any doubt, is related to the previous point. Most unfortunately, we do miss large numbers of direct CO₂ measurements from streams at very high altitude.

The site with the highest altitude contained in our model is from Qu et al. (2017, Scientific Reports) from the Tibetan Plateau – at 4935 m a.s.l. Therefore, we cut our data at 4900 m a.s.l.; less than 1% of the streams drain catchments higher than that. Please, see our comments below related to possible freezing and seasonality.

3. Range of discharge samples: The discharge data look strange. In Fig S3-C, they range from about $1 \text{ m}^3 \text{ s}^{-1}$ (ln discharge = 0) to less than a few milliliters per second (ln discharge = -12). This would mean that a significant part of the CO₂ data are from stream sections that are not covered by global data sets. The Swiss sample reanges from 0.02 – $2 \text{ m}^3 \text{ s}^{-1}$ which would translate to a lower ln limit of about -4. Is Fig. S3-C correct?

AUTHORS: Overall for this study, we decided to exclude streams with discharge (Q) higher than $2.26 \text{ m}^3 \text{ s}^{-1}$ to comply with the boundaries linked our hydrological scaling relationships. This confines our estimates to small mountain headwater and makes our estimate even more conservative, which we deem to be important. For the CO₂ model, we were able to compile data also from very small streams with Q estimated from a raster layer within the GloRiC database that allowed us to include very low Q values. However, given that these are estimates, we decided to remove streams with very low Q values from the model (as pointed out by this reviewer). We set the lower cut-off at $\ln Q = -10$, which gives us a median Q of $0.0596 \text{ m}^3 \text{ s}^{-1}$, with range from 0.000075 to $1.97 \text{ m}^3 \text{ s}^{-1}$. For technical reasons of data availability, we had to set the lower bound of the global data set at a Q of $0.001 \text{ m}^3 \text{ s}^{-1}$, so well above the lower model bound. At the same time, only 1.7 % of the global streams have a Q higher than the upper Q bound of the model ($1.97 \text{ m}^3 \text{ s}^{-1}$); given the distribution of the Q

data, we find this difference still reasonable for an extrapolation.

4. Range of soil data. Most of the soil organic carbon (SOC) data are centered within the high range (10-40%, $\ln[\text{SOC}] = 4.5\text{-}6 \text{ g kg}^{-1}$). This is problematic because in general SOC decreases with altitude and reflects the type of vegetation cover.

AUTHORS: We are grateful for this comment and do understand the reviewer's concern. As brought up by reviewer 1, high-altitude catchments can indeed also include remarkable stocks of organic soils and even peat. We retrieved soil data from the SoilGrids 1 km database (Hengl et al. 2014), which is typically used for global assessments. Comparing these data with others, it could be that they are slightly higher than what would be measured. For instance, the Grand et al. (2016) paper in PlosOne reports measured SOC contents of ca. 15% for the top layer, whereas the SoilGrids 1 km would give us ca. 25% — not bad though for a global extrapolation effort. This study also shows the rapid SOC decline with depth in an Alpine catchment. For this reason, we used the topsoil SOC (s11) from SoilGrids 1 km. The SOC data that we used in our CO₂ model have a median value of 21.6%, and range from 0.2 to 36.6%. Thus, we do cover a broad range over three orders of magnitude that, with all respect, we find overall reasonable for upscaling.

5. Model performance: The model narrows the two orders of magnitude in observed CO₂ concentration data (3 - 400 micromolar) down to a factor of 5 in the predicted range (Figure S3E). This raises questions, whether the model approach is really useful: Global estimates would probably not change significantly if just the median value and percentiles for the CO₂ concentration were used in the global calculations.

AUTHORS: This is an important point. Would we simply use the model per se (!) to predict CO₂ concentrations, we would overestimate, to some extent at least, low values and underestimate elevated values. The point is that we “simulated” (Monte Carlo) CO₂ concentrations from 10'000 iterations using the residuals from the CO₂ model. Therefore, the likelihood to catch low or high values, overestimated or underestimated, respectively, is relatively low. We have better explained this in the Methods and to some extent also in the main text of the manuscript.

The authors should critically review their database and expand it or discuss the limitations and uncertainties of the model more explicitly. One obvious way to expand the database is the use of CO₂ values obtained from alkalinity and pH. Although the quality of wet-chemistry data at low pH, low alkalinity and high DOC is questionable, the large remaining set of data will significantly improve the statistical power of the model outlined in Fig S3. Two additional governing factors require attention:

AUTHORS: Thanks for the suggestion to expand the database. May we here refer to our responses above as related to the use of pH and alkalinity to calculate CO₂ concentration.

6. Geology: An extended database of CO₂ in rivers would also allow the correlation with geology. Weathering of carbonate rock is a clear feature of the Swiss data (Fig S6). There is a need for more coverage of terrains with igneous rocks or basalt. (See the GLiM database by Hartmann and Moorsdorf, 2012 G3 13, Q12004.)

AUTHORS: This is a good suggestion for which are grateful. The Glim database has a spatial resolution of 0.5° and therefore hardly suitable for our extrapolation. Furthermore, the GliM database would not provide continuous data to be include in the model. Please note that 36% of the catchments used in our model are overwhelmingly underlain by carbonate rock, 21% by siliciclastic rock, and 20% by metamorphic rock. We have now added this information to the main text.

7. Seasonality: Outside the tropics, seasonality in river flow increases dramatically with altitude. Freezing temperatures and snow cover will reduce stream flow in the cold season, so that typical field observations only cover half of the year.

AUTHORS: This is an important point, which we have now better discussed in our manuscript. We are grateful for this. Depending on the latitude, streams can freeze partially with altitude during winter. During that period, streamflow is nourished by groundwater that has a temperature above the ambient air temperature. Therefore, these streams typically do not freeze but they can be covered by snow. However, depending on groundwater upwelling and exposure of the terrain, reaches with no snow cover emerge. These are hotspots for CO₂ outgassing to the atmosphere and that had entered the snow-covered channel upstream.

We do acknowledge that in any upscaling effort, such small-scale spatial and temporal heterogeneities are difficult to account for. Flow intermittency in streams drains arid and semi-arid, or karst catchments, is an analogy to our situation and one that has never been considered in any of the published scaling efforts.

The different weak spots in the model for predicting CO₂ concentrations in mountain rivers lead to the key question: Are the high CO₂ emission rates predicted in this paper for the world's mountain streams plausible?

AUTHORS: As mentioned above and as recognized by this reviewer, there is currently no "the best way" for upscaling. We combine novel k_{600} scaling relationships with direct CO₂ measurements, computational methods (incl. uncertainty) that do not rely on data aggregation, and are therefore convinced that our effort provides estimates of CO₂ emissions from mountain streams at least as reliable (if not more) as previously published fluxes. This notion is certainly corroborated by reported CO₂ measurements (calculated and directly measured) from mountain regions and that were not included in our model. We have referred to these already in the original submission.

8. Groundwater inflow: As a partial answer, the authors perform a validation exercise for the 4000 Swiss streams. They calculate the CO₂ concentration in the groundwater needed to support the evasion rates and compare those with "literature data" (Figure S5). Table S2 reveals that these literature data are all personal communications without any additional information. For a proper validation these values should be cleaned up (nobody measures CO₂ in water to 5 digits precision), the data need to be georeferenced and a comparison between the CO₂ mass balance and the measurements should be given for the different sites.

AUTHORS: This is a valid point for which we are grateful. We have now changed the respective SI Table accordingly and we removed the data from Laos, Czech Republic but also the sample from lowland Switzerland. Few are groundwater CO₂ data from mountain settings. Therefore, we added those originally. In the meantime, we were able to measure pCO₂ in a few springs (that is, groundwater) draining into our study streams in Switzerland. These values were now added to the data sets, and the overall picture remains unchanged, that is, observed CO₂ concentrations in the groundwater are well within the range of what would be expected to fully satisfy the CO₂ evasion flux from streams by groundwater CO₂ deliveries (see response to reviewer #1 for more details on the topic).

9. For the global estimate of a CO₂ emission by mountain streams of almost 200 Tg per year or 10-30% of the global emission rates, 650 – 1800 Tg per year) this study should show more convincingly where the carbon comes from. High mountain terrains with their high k_{600} values exhibit low primary production and often short seasons for soil respiration. The authors should address this question more clearly.

AUTHORS: We are grateful for this point. The previous comment relates to our mass balance calculations across 4000 Swiss streams showing that groundwater is a potential delivery route of CO₂ to the streams. Our stable isotope analyses show that CO₂ from soil respiration is indeed a source and that “lithogenic” sources may become more important with increasing altitude. Given the fact that the areal CO₂ fluxes are unexpectedly high, we deemed these analyses critical. Here we want to stress that the traditional CO₂ upscaling papers typically do not blend these various approaches as we do. We consider this as a major asset to our study.

Please note that elevated k_{600} does not affect the primary production in streams. It would simply outgas the oxygen produced by the primary producers more rapidly if its partial pressure is higher than in the atmosphere. Primary production can be substantial in mountain streams when discharge and related near-bottom hydraulics allow; the contributions (direct and indirect) from in-stream primary production (or was heterotrophic respiration meant by this reviewer?) to CO₂ fluxes are currently unknown. We believe that this discussion is well beyond the scope of our study.

Minor comments:

line 32 “groundwater CO₂ deliveries from rock weathering and soil respiration” - this is a bit a side track, because it has been known since the work of Garrels, Berner and others in the 1980ies that rock weathering transfers atmospheric CO₂ to the hydrosphere. A detailed discussion of weathering versus soil respiration would call for an expanded model with geological information (see remark 6)

AUTHORS: This is a relevant point and we would like to invite this reviewer to read our response to reviewer #1 where we do discuss this in some detail.

line 33: Not clear what the 1% uncertainty refers to (192 +- 1.9 Tg C yr⁻¹). In the light of the many model limitations outlined above such a high precision seems questionable. The 5 and 95% error bands in Figures 2 E – H seem to tell a different story.

AUTHORS: Thanks for pointing this out. We have propagated the error associated with k_{CO_2} , CO₂ concentration, stream width and flow velocity; as suggested by reviewer #3 (please, see below) we have now also propagated the error related to the conversion of air to streamwater temperature. For each stream, we perturbed the relationships (e.g., for CO₂ concentration) 10,000 times by randomly extracting error approximations from their corresponding residual probability distribution. We thereby created for each Monte Carlo simulation a random extraction of the CO₂ concentrations, stream width, flow velocity, streamwater temperature and k_{CO_2} values for all streams used in the simulation, and finally 10,000 estimates of CO₂ emission fluxes. For each iteration, we derived a total flux by summing up the fluxes from all streams accounting for their contributing area. We thereby obtained 10,000 total flux estimates, from which we extracted the mean CO₂ evasion flux estimate as well as the 5th and 95th percentiles as confidence intervals.

It is clear that this approach yields relatively small uncertainties. This is obvious as summing up areal fluxes with large uncertainties, one ends up with a total sum with a small uncertainty, because if errors are independent, they average out. This is well known as the central limit theorem in data science.

Our revised version now contains an alternative approach that considers error dependency between streams, which, not unexpected, yields a much larger uncertainty. Reality is likely between both approaches (as based on the assumption of error dependency versus independency). The new Method section contains an extensive description of the uncertainty calculation.

Please, see also our comment below to reviewer #3 on the same point.

lines 137 138, realistic precision for CO₂ values needed (3 significant digits is quite demanding). This paragraph refers to data from different parts of the world, but neither Figure S5 nor Table S2 provide georeferenced information. It is strange that the data calculated for 4000 Swiss river segments are compare to a random sample of data from the Chez Republic and Laos.

AUTHORS: Thanks for pointing us to this. We have changed this accordingly, also as a response to the reviewer's comment above.

line 146 “we were not able to include alkalinity as a potential sink for CO₂ in the mass balance”. Why not? By the way, the pool of carbonate alkalinity could also act as a source if stripped with air bubbles.

AUTHORS: Thank you for this comment. Including alkalinity would have added value to our study. Unfortunately, we do not have alkalinity data to be accounted for as a potential CO₂ sink. We have acknowledged this in the text.

line 273 – formatting ref 5

AUTHORS: We have changed this. Thanks!

line 279 ref 8 needs updated page numbers

AUTHORS: We have changed this. Thanks!

line 311 clean up citation Butmann et al.

AUTHORS: We have changed this. Thanks!

line 318 update ref 23.

AUTHORS: We have changed this. Thanks!

line 368 legend Figure 1 should mention that these are modelled distributions. Not clear what “multiple” stable isotopic analyses means. There is only one isotope measured.

AUTHORS: Thanks for this comment. We have detailed this and changed the sentence accordingly.

Figure 1 and other Figures in the supporting information: The exponents d-1 in Figure A and m-2 yr-1 in Figure C are not formatted correctly.

AUTHORS: We have changed this. Thanks!

Figure S5 There are no literature values in this supplement – only personal information.

AUTHORS: We have changed this. Thanks!

Reviewer #3 (Remarks to the Author):

Overview – Horgby et al present a compelling analysis that improves upon many of the recent estimates of stream and river CO₂ emissions. In particular, their work focuses on high elevation systems that are based upon recent findings from a 2 year effort in the Swiss Alps. Their works supports the hypothesis that groundwater and soil respiration contribute significant inorganic carbon to small mountainous streams, that that the physical environment of high slope and turbulent conditions creates conditions where evasion is high resulting in low measured CO₂ concentrations. Using a simple mass balance approach, the authors support the potential source of groundwater carbon dioxide within these systems. This manuscript is well done, and the analyses are complete. However, it is a shame that much of the analysis is lost within the supporting information and may never get highlighted as a result of the short format of Nature Communications.

AUTHORS: We are most grateful to this reviewer recognizing the potential relevance of our work. Our original submission was ca 2600 words in length. Given that 5000 words are allowed, we have now moved several parts from the SI to the main text and better highlighted and discussed some of our findings. We do hope that the revised version makes an even more compelling case.

Major Comments:

There are two aspects of this work that I believe warrant more explicit discussion for this to be published, and I am sure that these can be handled well by the authors. Within the manuscript as written, there is very limited discussion on the temporal nature of CO₂ concentrations in streams. This is very important when attempting to scale to annual estimates. This reviewer acknowledges that datasets are not yet available at a scale to properly constrain temporal dynamics globally, there should be more explicit discussion – perhaps based on the findings from the continuous monitoring within the Alps, how deviation in high elevation concentrations and subsurface CO₂ production may influence these global estimates. Along these lines, is it possible to provide a context for what 2000m means in the tropics vs. northern latitudes? If a system freezes – what is the impact on the annual emissions. If this was discussed explicitly, this reviewer did not see it. Do the authors assume no emissions for a portion of the year when frozen? Does precipitation drive emissions at all? It was surprising that this did not factor into the simple linear model to predict CO₂ concentrations on its own.

AUTHORS: This is an important point indeed for which we are most grateful. While most upscaling approaches have ignored the temporal variability of CO₂ evasion fluxes from streams, we are now discussing this important issue at least. As a matter of fact, the PI's lab was among the first to highlight the relevance of seasonal and diurnal variability of CO₂ evasion fluxes from streams (e.g., Peter et al, 2014 JGR-BG; Schelker et al. 2016 L&O).

To address the question of temporal variability, we have analysed the time series of CO₂ evasion fluxes (at 10 min intervals) from eight of our twelve streams in the Swiss Alps. We found that the median fluxes calculated from these time series reflect very well the predicted CO₂ flux from our model approach; in fact, the slope of the linear regression relating measured and computed median fluxes from these streams is statistically not significant from 1.

We had this analysis in the SI of our initial submission, and decided to better highlight it now in the revision by discussing it directly in the main manuscript.

The change of the 0° isotherm is well known and could possibly be considered. However, as discussed above (please, see reviewer #2), streams draining high-alpine catchments do not necessarily freeze/snow cover over their entire length. Because these systems are almost

exclusively groundwater fed during winter, they can easily and often have open reaches owing to the elevated temperature of groundwater relatively to the ambient air temperature. Such open reaches are hotspots that evade the CO₂ collected by the stream from the groundwater further upstream but where the CO₂ could not be vented because of the snow cover. We observe this very often during our winter surveys when large “bubbles” exit snow-covered reaches.

Therefore, it is far from trivial to account for such small-scale temporal and spatial heterogeneities. Frankly, I wish we could and I am certain this will be one of many exciting new research questions to be addressed in the future.

Our data set of 1,872,874 streams does not include any stream above 4,938 m a.s.l. (this is the highest site contained in our CO₂ model). Within this large sample, 97'459 streams are located between 3,500 and 4,938 m a.s.l., and owing to their typically very low pCO₂, these streams were typically CO₂ sinks (median areal flux: -0.55 kg C m⁻² yr⁻¹).

In this context, we want to reiterate that the global flux that we present is a net flux taking into account CO₂ fluxes from the atmosphere to the streams as well. The latter make sense as they most likely originate from mineral dissolution — an exciting new field just now being discovered in the high north.

Given the very large potential range in the predictors used to model emissions, it is surprising that the estimated error derived from the Monte Carlo for flux is only 1% at 1.9 Tg-C. In fact this level of precision is somewhat suspect. The authors could provide some additional clarity on how this was developed and reduced and what that might mean for interpretations.

***AUTHORS:** We are most grateful for this important comment, which prompted us to detail the description of the uncertainty calculations and to discuss our approach in the main text. Reason for this is that we chose an alternative to the typical uncertainty calculations used for CO₂ upscaling (e.g. Raymond et al. 2013, Lauerwald et al. 2015). It was our intention to propagate the potential error associated with k_{CO_2} , CO₂ concentration, stream width and flow velocity; as suggested by this reviewer (please, see below) we have now also propagated the error related to the conversion of air to streamwater temperature. In summary, our flux computation includes and propagates more error sources than any previous study. For each stream, we perturbed the relationships (e.g., for CO₂ concentration) 10,000 times by randomly extracting error approximations from their corresponding residual probability distribution. We thereby created for each Monte Carlo simulation a random extraction of the CO₂ concentrations, stream width, flow velocity, streamwater temperature and k_{CO_2} values for all streams used in the simulation, and finally 10,000 estimates of CO₂ emission fluxes. For each iteration, we derived a total flux by summing up the fluxes from all streams accounting for their contributing area. We thereby obtained 10,000 total flux estimates, from which we extracted the mean CO₂ evasion flux estimate as well as the 5th and 95th percentiles as confidence intervals.*

It is clear that using this approach we get a relatively small uncertainty. This is obvious as summing (largely uncertain) areal fluxes up, one ends up with a total sum with small uncertainty, because if errors are independent, they average out. In data science, this is well known as the central limit theorem.

Our revised version now contains an alternative approach that considers error dependency between streams. Not unexpected, this yields a larger uncertainty. The real uncertainty is likely between both approaches — that is, error dependency and independency. The new Method section contains an extensive description of the uncertainty calculation.

The only other aspect that should be addressed is the potential impact of the available datasets on stream width and hence surface area. This reviewer agrees with the authors that this work could be considered conservative. In fact the spatial datasets used for the global analysis appears to only work at a resolution of 10m or greater? It is recommended that the authors discuss the potential loss of streams below these thresholds if possible in more detail. The authors could utilize the cited paper Allen et al. 2018 – Nature Comm. (<https://doi.org/10.1038/s41467-018-02991-w>) This citation details a potential model for capturing very small streams in an area calculation.

AUTHORS: Thank you very much for this comment. The Allen et al., 2018 is an outstanding paper indeed. However, we did not use their stream width model 2018 because it requires, besides streamflow, additional estimates of hydraulic resistance and natural variability of channel geometry, which we cannot easily reproduce at a global scale within the scope of this study. Nevertheless, we would like to argue that we still captured small streams in our study. The GloRiC database provides streams and rivers at all locations where discharge exceeds $0.1 \text{ m}^3 \text{ s}^{-1}$, or upstream catchment area exceeds 10 km^2 , or both. In many mountain areas, the 10 km^2 condition applies first.

According to our hydraulic scaling relationship (please, see equation 1 in the revised Methods), the maximum channel width calculated is 10.2 m; because we restricted Q ($<2.26 \text{ m}^3 \text{ s}^{-1}$), our data set does not allow wider channels. Our scaling approach yielded minimum channels width of 0.32 m — coincidence or not — the same ($0.32 \pm 0.07 \text{ m}$) as reported as most abundant by Allen et al. 2018 using a lognormal statistical distribution

Minor comments:

NO REFERENCE 5?

AUTHORS: Thank you very much for your detailed look. We have fixed this now.

34 – relatively...

AUTHORS: Changed to “relatively low”.

35 – hitherto... awkward

AUTHORS: “hitherto unrecognized contributors” changed to “significant contributors”.

69 – used per/mil not percent

AUTHORS: This is correct. The unit for relief roughness is per/mil. As explained in the Methods “[...] relief roughness was calculated as the difference in a pixel’s maximum and minimum elevation divided by half the pixel length.”

102-106 – can the authors bring in more description of how the error of using the relatively weak linear model propagates into the estimates of CO₂ concentration here?

AUTHORS: Please, see our comment to this reviewer’s comment above on the uncertainty calculation. As mentioned above, we have now added information on the uncertainty calculation in the main text and refer to a fully re-written Method section.

122 – is geopredictors a real term?

AUTHORS: This term is often used in Earth system models.

226 – change culminating to summing...

AUTHORS: “culminating” changed to “summing”.

Methods –

When converting air temperature to water temperature, was there any analysis that suggest these systems that are governed by turbulence across adhere to the cited equation? Also - was this component of the analysis included within the Monte Carlo assessment?

AUTHORS: We are most grateful for this comment. We have now implemented this component in our Monte Carlo simulations. Please, see also above.

112 – converted...

AUTHORS: Thank you for this comment. We removed “Besides elevation” for clarity.

148- can you provide a figure for how the miller tans approach separated the potential sources? This can be added to the supporting information with. It would appear that there is a 10 per mil range in the swiss alps dataset, are there additional datasets that can contribute here?

AUTHORS: Please see response to reviewer #1. Based on the critical view of that reviewer, we decided to remove the Miller-Tans analyses and, instead, we use Keeling plots to estimate DIC sources (updated Figure 1E).

Reviewers' comments:

Reviewer #1 (Remarks to the Author):

Review of Horgby et al.

First of all, I want to commend the authors on their very detailed and constructive author replies (and for clearing up my confusion re. GloRiC and GloRiCh datasets). While my own evaluation focussed on other aspects, I do agree with the other reviewer comments and the reservations on the global extrapolation – however, I feel the authors are sufficiently transparent in the revised version on the shortcomings and pitfalls so that readers can evaluate for themselves how robust these first numbers are. Undoubtedly they will be revised in the future, but such a first extrapolation is still useful to point out the potential importance of a process and to stimulate further work. Hence despite some reservations I don't have a problem seeing the flux extrapolation published in a high-ranking journal, even if in hindsight the flux may turn out to be strongly over- or underestimated, this is an exercise that should stimulate others to improve these estimates.

On the other hand, I do still have an issue with the way the stable isotope data are handled in the revised version. I don't like to spoil things but my feeling is that even this downscaled interpretation of the stable isotope data is fundamentally incorrect and should be removed from the manuscript. I have tried to provide some more detailed arguments for my point of view below, I'm hoping these are clear and can convince the authors of a few pitfalls.

The authors cite a number of papers to indicate that the Keeling plot approach is widely accepted ("well rooted") in ocean, lake and river studies. This is unfortunately exactly the problem I have: it is an elegant approach but all too often inappropriately applied to data where the underlying assumptions are not met. It is deceptively elegant in that it often results in seemingly consistent results, but often these are not really meaningful. I have seen this approach applied to data collected along salinity gradients, in sediment porewater profiles, etc – all situations where it should not be used. The Mortazavi & Chanton study the authors refer to (and the Campeau et al. references) are good examples of the same problem. It is an issue that I have run into regularly when reviewing manuscripts and I have had long discussions on this with authors and colleagues, and noticed how hard it is to convince people to abandon this approach except in clear cases where the scenario it was developed for is applicable: a system where a given background pool of an element with a given isotope composition changes over time due to the addition of a source with a different isotope composition – and no other processes at play i.e. no concurrent losses, no isotope fractionation etc. One positive example is the Karlson et al. 2007 L&O study the authors refer to – here it is applied to short-term incubations where the initial DIC pool changes due to respiratory DIC inputs – if we assume no other processes are significant during the incubation period, the approach is fine. In more complex systems however, such as estuarine mixing zones or when using samples from across different sites in rivers or lakes, ocean depth profiles etc – this no longer holds: DIC concentrations and $\delta^{13}\text{C}$ values are not merely the results of an addition of DIC to a background pool, but also influenced by outgassing, photosynthesis, carbonate precipitation/dissolution, mixing of water sources with different [DIC] and $\delta^{13}\text{C}$ -DIC, etc.

To illustrate the caveats with Keeling or Miller-Tans plots, perhaps consider this thought experiment: consider two water sources (e.g. a freshwater and marine end-member, or a more appropriate comparison in the context of this study: a river surface water and a groundwater source), each with distinct DIC concentrations and $\delta^{13}\text{C}$ -DIC values. Then plot the theoretical data collected along a mixing gradient (conservative mixing, i.e. no additional sources or sinks), e.g. surface waters with different contributions of the groundwater source. Applying the Keeling or Miller-Tans approach to such a dataset would give you great-looking plots and a sometimes seemingly realistic output (intercept) for the $\delta^{13}\text{C}$ of the 'source' of DIC, while in fact there is not even a 'source' to consider. Play around with the end-member values and see what happens to your intercept. Keeping this example in mind, and factoring in CO_2 gas exchange, photosynthesis, respiration, carbonate dissolution/precipitation and isotope fractionation associated with some of these processes – it should become evident that taking data collected at different sites and dates across river basins should not be expected to allow characterizing the $\delta^{13}\text{C}$ value of the 'source' of newly added DIC.

Furthermore, whether one uses the Keeling plot method or Miller-Tans plots is not really relevant – both are based on the same assumptions but use a different expression and graphical approach to solve the same underlying equation. They may or may not lead to different results – depending on the range of DIC (or CO₂, or whatever element/matrix one is looking at) concentrations, and the regression methods used. There are good discussions on the intricacies of this comparison in numerous papers, e.g.

Chen et al. (2017) Inter-comparison of three models for δ¹³C of respiration with four regression approaches. *Agricultural and Forest Meteorology* 247: 229-239.

In short, I feel it's important not to fall into the same trap as earlier papers have done and take out the Keeling plot interpretation of the data – it is inherently an incorrect approach to analyze the available data. While it may be possible that the conclusions drawn are not far off reality, it would take a more complex analyses to investigate this, and keeping the current analysis in the ms will only lead to others picking up and applying the same approach, taking it as a “well-established” approach in aquatic sciences.

In addition, the authors' reply on the use of δ¹³C to distinguish biogenic and lithogenic C is not reassuring (points 3 and 4 in their response): first, CO₂atm is unlikely to be directly involved in carbonate weathering as the latter will mostly occur belowground – the atmospheric CO₂ 'sink' is mostly because weathering consumes CO₂ that would otherwise evade to the atmosphere. The actual CO₂ used is more likely to be derived from mineralization, or of 'geogenic' (used here in my interpretation of the term) origin e.g. in volcanically active mountain regions. Secondly, the effect of gas exchange / CO₂ outgassing cannot be underestimated here – δ¹³C-DIC can shift up to 10‰ between a spring and short distances downstream due to intensive outgassing.

Finally, the authors' final numbers have changed from 192 ± 1.9 Tg C y⁻¹ in the original version to 167 ± 1.5 Tg C y⁻¹ in the revised version (abstract, L 32). It is not immediately clear to me which re-analysis resulted in this 13% reduction, this should be clarified explicitly. Assuming the underlying data have not changed, this should be a good illustration that the uncertainties provided are extremely optimistic.

Reviewer #2 (Remarks to the Author):

The manuscript and the SI have been carefully revised and corrected based on the different comments of three reviewers. The explanation of the workflow is now improved, issues like seasonality, geology, uncertainty, data sources and range of data are now addressed although not every critical aspect was fully resolved. In some cases the authors have chosen to argue their case instead of improving the paper. Examples:

* The authors make a point why they rely on sensor measurements - "owing to the very large uncertainty associated with the CO₂ estimates from alkalinity and pH..." (rev#2, comment 1). In this general form, the argument is not correct. There are known factors limiting the alkalinity-pH method but in the majority of cases the method yields results that are competitive in accuracy with CO₂ sensors, which have their own limitations such as calibration problems and sensor drift. Nonetheless, I understand that expanding the database would have resulted in a significant additional workload, that may not be warranted at this point.

* It remains a puzzle, how CO₂ production in alpine streams and transfer from their catchments could be large enough to support the "unexptected large evasion fluxes". I regret the authors' conclusion that a more detailed discussion of potential sources is "well beyond the scope of our study". Because it is difficult to see why CO₂ sources in the mountains should be stronger than in the lowlands, I expect that a more detailed seasonal analysis with improved global coverage will result in a significantly lower global CO₂ flux from mountain streams.

In summary, the authors did a careful job in revising their and I recommend publication in this form. This study will trigger additional research which might resolve the puzzling magnitude of this global extrapolation.

Reviewer #3 (Remarks to the Author):

Dear Authors,

This reviewer is satisfied with the comments as they were addressed, and the manuscript in its current form is adequate for publication. This reviewer appreciates both the tone of discourse in the response to reviewer comment letter, as well as the detail and justification for the improved methods.

Reviewers' comments:

Reviewer #1 (Remarks to the Author):

Review of Horgby et al.

First of all, I want to commend the authors on their very detailed and constructive author replies (and for clearing up my confusion re. GloRiC and GloRiCh datasets). While my own evaluation focussed on other aspects, I do agree with the other reviewer comments and the reservations on the global extrapolation – however, I feel the authors are sufficiently transparent in the revised version on the shortcomings and pitfalls so that readers can evaluate for themselves how robust these first numbers are. Undoubtedly they will be revised in the future, but such a first extrapolation is still useful to point out the potential importance of a process and to stimulate further work. Hence despite some reservations I don't have a problem seeing the flux extrapolation published in a high-ranking journal, even if in hindsight the flux may turn out to be strongly over- or underestimated, this is an exercise that should stimulate others to improve these estimates.

AUTHORS: We are grateful for these insights. We do agree with this reviewer that global extrapolations are to be interpreted with caution. This is best illustrated by the Drake et al. (2018) paper in Limnology and Oceanography Letters, where they show how global estimates of CO₂ emissions from inland waters have evolved over the last decade or so — with studies published in Nature, Nature Geoscience, Limnology and Oceanography etc. Our study makes a strong point that we have not yet reached consensus with these estimates as a proper estimation of CO₂ emissions from mountain streams has been factually neglected so far. Therefore, our study is an important further step to the effort of reaching well-constrained CO₂ fluxes. We have acknowledged this in the concluding paragraph of the manuscript.

In this context, we want to re-iterate what we mentioned previously, namely that our extrapolation includes carefully thought-through considerations and computational steps, which makes it probably more robust than previous efforts. While this information may be of relevance to a rebuttal letter, it is of no relevance for the manuscript itself. Scholars will make their own opinion.

On the other hand, I do still have an issue with the way the stable isotope data are handled in the revised version. I don't like to spoil things but my feeling is that even this downscaled interpretation of the stable isotope data is fundamentally incorrect and should be removed from the manuscript. I have tried to provide some more detailed arguments for my point of view below, I'm hoping these are clear and can convince the authors of a few pitfalls.

The authors cite a number of papers to indicate that the Keeling plot approach is widely accepted (“well rooted”) in ocean, lake and river studies. This is unfortunately exactly the problem I have: it is an elegant approach but all too often inappropriately applied to data where the underlying assumptions are not met. It is deceptively elegant in that it often results in seemingly consistent results, but often these are not really meaningful. I have seen this approach applied to data collected along salinity gradients, in sediment porewater profiles, etc – all situations where it should not be used. The Mortazavi & Chanton study the authors refer to (and the Campeau et al. references) are good examples of the same problem. It is an issue that I have run into regularly when reviewing manuscripts and I have had long discussions on this with authors and colleagues, and noticed how hard it is to convince people to abandon this approach except in clear cases where the scenario it was developed for is applicable: a system where a given background pool of an element with a given isotope composition changes over time due to the addition of a source

with a different isotope composition –and no other processes at play i.e. no concurrent losses, no isotope fractionation etc. One positive example is the Karlson et al. 2007 L&O study the authors refer to – here it is applied to short-term incubations where the initial DIC pool changes due to respiratory DIC inputs – if we assume no other processes are significant during the incubation period, the approach is fine. In more complex systems however, such as estuarine mixing zones or when using samples from across different sites in rivers or lakes, ocean depth profiles etc – this no longer holds: DIC concentrations and $\delta^{13}\text{C}$ values are not merely the results of an addition of DIC to a background pool, but also influenced by outgassing, photosynthesis, carbonate precipitation/dissolution, mixing of water sources with different [DIC] and $\delta^{13}\text{C}$ -DIC, etc. To illustrate the caveats with Keeling or Miller-Tans plots, perhaps consider this thought experiment: consider two water sources (e.g. a freshwater and marine end-member, or a more appropriate comparison in the context of this study: a river surface water and a groundwater source), each with distinct DIC concentrations and $\delta^{13}\text{C}$ -DIC values. Then plot the theoretical data collected along a mixing gradient (conservative mixing, i.e. no additional sources or sinks), e.g. surface waters with different contributions of the groundwater source. Applying the Keeling or Miller-Tans approach to such a dataset would give you great-looking plots and a sometimes seemingly realistic output (intercept) for the $\delta^{13}\text{C}$ of the ‘source’ of DIC, while in fact there is not even a ‘source’ to consider. Play around with the end-member values and see what happens to your intercept. Keeping this example in mind, and factoring in CO_2 gas exchange, photosynthesis, respiration, carbonate dissolution/precipitation and isotope fractionation associated with some of these processes – it should become evident that taking data collected at different sites and dates across river basins should not be expected to allow characterizing the $\delta^{13}\text{C}$ value of the ‘source’ of newly added DIC.

Furthermore, whether one uses the Keeling plot method or Miller-Tans plots is not really relevant – both are based on the same assumptions but use a different expression and graphical approach to solve the same underlying equation. They may or may not lead to different results – depending on the range of DIC (or CO_2 , or whatever element/matrix one is looking at) concentrations, and the regression methods used. There are good discussions on the intricacies of this comparison in numerous papers, e.g.

Chen et al. (2017) Inter-comparison of three models for $\delta^{13}\text{C}$ of respiration with four regression approaches. *Agricultural and Forest Meteorology* 247: 229-239.

In short, I feel it’s important not to fall into the same trap as earlier papers have done and take out the Keeling plot interpretation of the data – it is inherently an incorrect approach to analyze the available data. While it may be possible that the conclusions drawn are not far off reality, it would take a more complex analysis to investigate this, and keeping the current analysis in the ms will only lead to others picking up and applying the same approach, taking it as a “well-established” approach in aquatic sciences.

AUTHORS: We respectfully accept the view of this reviewer and do not want to further argue along this line with her/him. Therefore, we have now removed the Keeling analysis from the manuscript. Instead, we do present the raw $\delta^{13}\text{C}$ -DIC data (as box plots) for all our study streams in the Swiss Alps, sampled across seasons (N=134), and put these in the context of the overall variability of $\delta^{13}\text{C}$, ranging from carbonate rock to organic matter. While this approach is the simplest possible, it still indicates carbonate as a relevant lithogenic source of DIC to the streams. The take home message thus remains unchanged.

In this context, may we make the point that the focus of this manuscript lies on the CO_2 fluxes from mountain streams. We used a simple mass balance approach that suggests groundwater as a potential route for CO_2 delivery to the streams. On top of this, our stable isotope analysis serves as an additional piece in the mosaic to support our notion of an external source — mostly lithogenic in nature — of the DIC/ CO_2 to the mountain streams.

In addition, the authors' reply on the use of $\delta^{13}\text{C}$ to distinguish biogenic and lithogenic C is not reassuring (points 3 and 4 in their response): first, CO_2atm is unlikely to be directly involved in carbonate weathering as the latter will mostly occur belowground – the atmospheric CO_2 'sink' is mostly because weathering consumes CO_2 that would otherwise evade to the atmosphere. The actual CO_2 used is more likely to be derived from mineralization, or of 'geogenic' (used here in my interpretation of the term) origin e.g. in volcanically active mountain regions. Secondly, the effect of gas exchange / CO_2 outgassing cannot be underestimated here – $\delta^{13}\text{C-DIC}$ can shift up to 10‰ between a spring and short distances downstream due to intensive outgassing.

***AUTHORS:** We are grateful for this comment. The context of our alpine systems (different from tropical and "temperate" systems) is one of melt waters from glaciers and snow, that recharge groundwaters and surface water, one with poor soil development and vegetation, and one without any volcanic activity. In this context, atmospheric CO_2 is likely involved in the weathering of carbonates, depending of course on the water temperature, hence CO_2 solubility, and on its partial pressure in the atmosphere. As such, the contribution of atmospheric CO_2 may remain relatively weak but more important compared to CO_2 from soil respiration; certainly, this shall change once we get to below the tree line. Atmospheric CO_2 dissolved in groundwater will result in a sufficiently low pH too to start dissolving carbonate in rocks - our "geogenic" CO_2 contribution. This pathway seems relevant to us as we do have carbonate lithologies in our catchment, but no volcanic or deep sources of CO_2 (volcanic or metamorphic). For every mole of atmospheric CO_2 dissolved in the cold water, one mole of DIC can be created by reaction with carbonates in the infiltration zone of the fractured rocks and the physically eroded sediments. Both of these will, however, then give a DIC of "heavy" isotopic composition. These waters will then exfiltrate and feed into the streamwater, where degassing will take place because of warming. As mentioned previously and in the manuscript, degassing will change the $\delta^{13}\text{C-DIC}$ values, particularly if degassing is substantial. The change in isotopic composition with this type of degassing from waters that have been "geogenically" charged with CO_2/DIC , will cause an increase in the $\delta^{13}\text{C-DIC}$ values. Please, note that this is part of our discussion.*

Finally, the authors' final numbers have changed from $192 \pm 1.9 \text{ Tg C y}^{-1}$ in the original version to $167 \pm 1.5 \text{ Tg C y}^{-1}$ in the revised version (abstract, L 32). It is not immediately clear to me which re-analysis resulted in this 13% reduction, this should be clarified explicitly. Assuming the underlying data have not changed, this should be a good illustration that the uncertainties provided are extremely optimistic.

***AUTHORS:** We are grateful for this comment. Clearly, the difference emanates from the fact that we modified the CO_2 prediction model, according to suggestions from Reviewer #2's in the last round of reviews: "3. Range of discharge samples [...]". As previously responded, we decided to exclude streams with discharge (Q) higher than $2.26 \text{ m}^3 \text{ s}^{-1}$ to comply with the boundaries linked our hydrological scaling relationships. However, in line with above mentioned remark from reviewer #2, we decided to also include a lower discharge boundary in the CO_2 prediction model ($Q > 0.01 \text{ m}^3 \text{ s}^{-1}$). Overall, this has reduced the number of streams now included in the extrapolation and the CO_2 model as well. Therefore, the underlying data have changed and this reduction is not related to uncertainty. We are sorry that this was not obvious from the last rebuttal.*

Reviewer #2 (Remarks to the Author):

The manuscript and the SI have been carefully revised and corrected based on the different

comments of three reviewers. The explanation of the workflow is now improved, issues like seasonality, geology, uncertainty, data sources and range of data are now addressed although not every critical aspect was fully resolved. In some cases the authors have chosen to argue their case instead of improving the paper. Examples:

AUTHORS: We are grateful that this reviewer has appreciated the revisions that we made. Please see our comments below.

* The authors make a point why they rely on sensor measurements - "owing to the very large uncertainty associated with the CO₂ estimates from alkalinity and pH..." (rev#2, comment 1). In this general form, the argument is not correct. There are known factors limiting the alkalinity-pH method but in the majority of cases the method yields results that are competitive in accuracy with CO₂ sensors, which have their own limitations such as calibration problems and sensor drift. Nonetheless, I understand that expanding the database would have resulted in a significant additional workload, that may not be warranted at this point.

AUTHORS: We used sensors that measure pCO₂ every ten minutes in the Swiss study streams. Besides these measurements, we built our predictive model on direct CO₂ measurements, not on those derived from chemical analyses this reviewer refers to. We are grateful for his/her comment and we toned down our statement accordingly.

* It remains a puzzle, how CO₂ production in alpine streams and transfer from their catchments could be large enough to support the "unexpected large evasion fluxes". I regret the authors' conclusion that a more detailed discussion of potential sources is "well beyond the scope of our study". Because it is difficult to see why CO₂ sources in the mountains should be stronger than in the lowlands, I expect that a more detailed seasonal analysis with improved global coverage will result in a significantly lower global CO₂ flux from mountain streams.

AUTHORS: We are grateful for this comment, which of course relates to the discussion that we are having with reviewer 1 (please, see above). We have presented two independent lines of evidence that the CO₂ comes from allochthonous sources: (i) mass balance calculations over 4000 streams suggesting that the CO₂ delivered via groundwater into the streams can satisfy in principle the evasion flux from the streams; (ii) stable isotope analysis suggesting a lithogenic origin (from carbonate rock).

We do not pretend that CO₂ sources in mountains are stronger than in lowlands, and we are sorry if this was understood as such by this reviewer. What we propose is that owing to the general topography and the fractured rock structure of the mountains, water residence time at landscape level is high (as cited by Kirchner, Nature Geoscience), and hence the probability for weathering as well. Furthermore, it is well established that owing to the high hydraulic permeability of alluvial sediments and high channel slopes, the hydrodynamic exchange between groundwater and the streamwater is high, hence streams are continuously and rapidly recharged by groundwater (obviously rich enough in CO₂ to satisfy the observed outgassing).

We do agree with this reviewer that "more seasonal analysis with improved global coverage" will better constrain the global CO₂ fluxes — however, not necessarily towards lower numbers. In fact, our comparison with the Swiss time series reveal a very good agreement between temporarily high-resolved data resolved and our model approach.

In summary, the authors did a careful job in revising their and I recommend publication in this form. This study will trigger additional research which might resolve the puzzling magnitude of

this global extrapolation.

Reviewer #3 (Remarks to the Author):

Dear Authors,

This reviewer is satisfied with the comments as they were addressed, and the manuscript in its current form is adequate for publication. This reviewer appreciates both the tone of discourse in the response to reviewer comment letter, as well as the detail and justification for the improved methods.

***AUTHORS:** We are most grateful to this reviewer for her/his help to improve our manuscript.*

REVIEWERS' COMMENTS:

Reviewer #1 (Remarks to the Author):

My main concern during the previous round of reviews was with the Keeling/Miller-Tans approach which in my opinion was not appropriate for this dataset, now that this has been removed from the ms I see no major flaws in the revised ms.

While somewhat early stage for a global upscaling, the topic is relevant and the data workflow is transparent, hence the reader can make their own interpretation on the robustness of the estimates, and having this work come out in a journal with high visibility will certainly stimulate further work and refinements in the future.